# EmergentTTS-Eval: Evaluating TTS Models on Complex Prosodic, Expressiveness, and Linguistic Challenges Using Model-as-a-Judge

**Ruskin Raj Manku    Yuzhi Tang    Xingjian Shi    Mu Li    Alex Smola**

Boson AI, Santa Clara, CA 95054
{ruskin, yuzhi, xingjian, mu, smola}@boson.ai

## Abstract

Text-to-Speech (TTS) benchmarks often fail to capture how well models handle nuanced and semantically complex text. Building on *EmergentTTS*, we introduce *EmergentTTS-Eval*, a comprehensive benchmark covering six challenging TTS scenarios: emotions, paralinguistics, foreign words, syntactic complexity, complex pronunciation (e.g. URLs, formulas), and questions. Crucially, our framework automates both test-case generation and evaluation, making the benchmark easily extensible. Starting from a small set of human-written seed prompts, we iteratively extend them using LLMs to target specific structural, phonetic and prosodic challenges, resulting in 1,645 diverse test cases. Moreover, we employ a model-as-a-judge approach, using a Large Audio Language Model (LALM) to assess the speech across multiple dimensions such as expressed emotion, prosodic, intonational, and pronunciation accuracy. We evaluate state-of-the-art open-source and proprietary TTS systems, such as 11Labs, Deepgram, and OpenAI's 4o-mini-TTS, on EmergentTTS-Eval, demonstrating its ability to reveal fine-grained performance differences. Results show that the model-as-a-judge approach offers robust TTS assessment and a high correlation with human preferences. We open source the evaluation code[1] and the dataset[2].

## 1 Introduction

Recent breakthroughs in generative modeling have led to significant advancements in Text-to-Speech (TTS) technology [6, 46, 18, 42, 14]. State-of-the-art proprietary systems now demonstrate remarkable naturalness and human-like quality when converting standard, well-formed text into spoken language. These systems are widely deployed in various applications, including virtual assistants [26], audiobooks [39, 27], navigation systems [20, 38], and accessibility tools [29, 25]. However, as TTS technology becomes more integrated into real-world use cases, systems increasingly encounter complex and diverse text prompts that go beyond conventional reading tasks, such as code switching, or rendering complex technical character sequences.

Conversely, evaluation methodologies for TTS systems have not kept pace with the growing complexity of use cases. Current benchmarks exhibit several limitations: they often use restricted text domains [44], the lack diversity in linguistic phenomena [40], and they rely on costly, non-reproducible human evaluations that may vary significantly across different listener cohorts. Even worse, code-switching in multiple languages requires extremely polyglot evaluators (or many specialized ones). Thus, for reasons of practicality, many evaluations focus on voice cloning alone.

---

[1] https://github.com/boson-ai/EmergentTTS-Eval-public
[2] https://huggingface.co/datasets/bosonai/EmergentTTS-Eval

Real-world TTS applications encounter numerous challenges that remain difficult for current systems. These include accurately reflecting human emotions and sounds [8]-for example, when narrating various types of books like fantasy or children's literature, TTS systems must realistically handle quoted dialogues and paralinguistic cues to keep listeners engaged. Another dimension involves more formal scenarios, such as syntactically complex text with nested clauses in legal and literary contexts, or scientific and academic texts containing special characters and equations that are difficult to pronounce. Additionally, there is a growing need for TTS systems to handle multilingual content [23, 11, 34] and properly intonate questions with contextually appropriate prosody [19], challenges that current evaluation frameworks fail to systematically address. An evaluation methodology is required that reliably captures TTS performance across all these scenarios, moving away from subjective human assessments of expressiveness and prosody.

To this end, we propose EmergentTTS-Eval, a comprehensive benchmark specifically designed to evaluate TTS performance across these challenging scenarios. Our benchmark covers six critical dimensions. Through an iterative refinement process, we are able to controllably generate increasingly more difficult utterances for TTS systems to synthesize. Furthermore, drawing parallel from the textual domain, where reward LLMs are widely used for judging output of other LLMs, we propose to use the model-as-judge paradigm for evaluating TTS systems. Our contributions are as follows:

- We create a benchmark with 1,645 samples for evaluating TTS systems across six challenging scenarios: Emotions, Paralinguistics, Syntactic Complexity, Questions, Foreign Words and Complex Pronunciation.
- We propose an iterative refinement strategy with LLMs that creates increasingly complex utterances for TTS, resulting in a multi-layered and diverse evaluation benchmark for evaluating all aspects of TTS performance.
- We are the first to use Large Audio Language Models (LALMs) as reward models for judging otherwise subjective dimensions of audio, like expressiveness, prosody, pausing, stress and pronunciation accuracy. We show its effectiveness through human correlation. The results are stable under the choice of judger LALMs.
- We evaluate leading open-source and closed-source TTS systems on our benchmark, showing how model-as-a-judge reveals finer-grained and systematic failures, and highlights the gap between closed-source and open-source models on specific aspects of speech generation.

## 2 Related Work

### 2.1 Text-to-Speech Model Evaluation Metrics

Traditional TTS evaluation rely on humans to provide a Mean Opinion Score (MOS) that is both costly and statistically noisy, due to its reliance on a changing pool of evaluators. Recent advances in automatic TTS evaluation typically rely on two metrics: the Word Error Rate (WER) [6, 46, 18, 14], as calculated by using an ASR model to convert the generated speech back into text and compare with the reference text; a speaker-similarity score (SIM) [6, 46, 18, 42], calculated by comparing the latent embeddings of generated vs. reference speech using an audio foundation model, such as WaveLM [15]. Recent works also explored the use of models to directly predict MOS (Sim-MOS) by training on datasets such as The Samsung Open MOS Dataset [24] and The VoiceMOS Challenge [16].

While these metrics serve to capture how natural or accurate a system sounds, their evaluation power is limited by the difficulty and expressivity of the voice dataset and cannot handle nuanced, context-sensitive phenomena such as emotional prosody or complex syntax. More recently, BASE-TTS [18] introduced an emergent abilities test suite that probes seven linguistically motivated phenomena, such as compound nouns, emotions, foreign words, paralinguistics, etc., using 20 hand-crafted prompts per category. Although BASE-TTS shifts the focus toward higher-order TTS capabilities, its dataset size is limited and reliance on human expert judgers is costly.

Our work addresses these limitations by automating test-generation and expanding category coverage. In particular, we create progressively harder stimuli at scale to differentiate between high-performing TTS systems. Our framework thus bridges the gap between traditional metric-based evaluation and nuanced, reproducible benchmarking.

## 2.2 Model-as-a-Judge For Text-to-Speech Model Evaluation

A key weakness in previous benchmarks is the need for human judges. Recent years have seen a growing trend of integrating audio encoders with LLMs. This has resulted in large audio-language models (LALM) that excel at a variety of audio comprehension tasks [15, 37, 30, 36, 12]. SALMONN [36] uses finetuned LALM to predict MOS, SIM and A/B testing scores. Wang et al. [41] extends this by finetuning an LALM to also generate open-ended qualitative feedback, covering noisiness, distortion, prosody, etc., alongside scores. This approach leverages the LLM's contextual knowledge to provide multi-dimensional evaluations more akin to a human expert. Chen et al. [10] compiled the first corpus of human-written TTS evaluations (with overall MOS plus detailed error annotations) and used it to train an audio-augmented GPT model. The resulting system can describe speech quality degradations and compare two samples in free-form language and outperforms prior state-of-the-art MOS prediction models. WavReward [17] employs a generalist reward model to score spoken dialogue quality across dimensions like clarity and expressiveness .

Our work not only use LALMs as judges but also to generate tests spanning categories of emergent TTS abilities. Our evaluation demonstrates that even out-of-the-box LALMs like Gemini-2.5 Pro are capable of evaluating emergent capabilities in SOTA TTS systems and produce A/B testing results that are highly-correlated with human preference.

## 3 EmergentTTS-Eval Benchmark

In this section, we describe how we construct the datasets in EmergentTTS-Eval, which covers 6 categories of challenging real-world TTS scenarios with varying levels of complexity. We also describe how the evaluation process is scaled with the help of Large Audio Language Model (LALM).

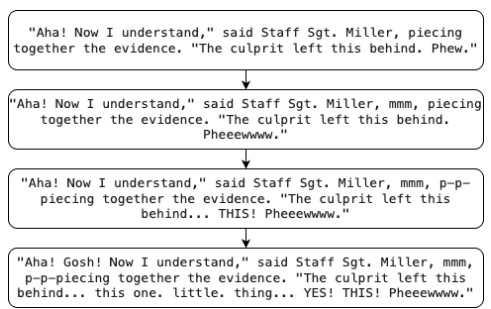

Figure 1: Paralinguistic example, refined and made more complex for TTS with increased number of cues.

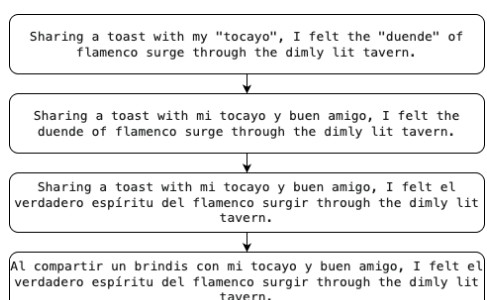

Figure 2: Foreign words example, refined and made more complex by adding idiomatically and prosodically rich foreign words.

## 3.1 Dataset Construction

We follow two key guidelines when constructing text prompts in EmergentTTS-Eval: (1) the dataset should encompass real-world challenges faced by TTS systems, and (2) it should exhibit varying levels of difficulty to enable fine-grained assessment of system capabilities. To this end, we begin with a diverse set of seed prompts and iteratively expand their scope (breadth) and complexity (depth). This reflects the approach of progressively increasing instruction difficulty used in instruction tuning [45]. Our seed prompts are derived from a collection of 140 human curated samples introduced in the BASE-TTS [18]. These samples span 7 challenging TTS categories commonly encountered in real-world scenarios: "Compound Nouns", "Emotions", "Foreign Words", "Paralinguistics", "Questions", "Syntactic Complexities", and "Punctuation", with 20 samples per category. Some prompts pose challenges for TTS systems because generating realistic speech requires a deep understanding of the text's semantic context. Consider the following text prompt from the "Emotions" category: `A profound sense of realization washed over Beal as he whispered, "You've been there for me all along, haven't you?  I never truly appreciated you until now."`. An effective TTS system should recognize the emotional context and appropriately render the quoted sentence as a whisper to reflect Beal's sentiment.

Although the BASE-TTS proposed set is of high quality, it is limited in its ability to explore the depth within each category and lacks broad diversity, as it was curated by a small group of individuals. However, complexity and diversity are essential for a robust evaluation benchmark, as they help assess challenging scenarios where system failures can significantly impact user experience. For example, we want to evaluate on real-world scenarios like sequential interrogative questions, sustained emotion synthesis with natural shift to contrasting emotions, multi-code switching, etc. In addition to the categories defined in BASE-TTS, we introduce a new category called "Complex Pronunciation", which contains prompts featuring unusual characters, numerals, and tongue-twisters. We exclude the "Compound-Nouns" category due to it's limited scope and the strong performance of current TTS systems according to manual assessment. We also drop the "Punctuations" category, as punctuation-related challenges are inherently addressed within other categories such as "Paralinguistics", "Syntactic Complexity", and "Complex Pronunciation".

To enrich the complexity of the text prompts and improving their diversity, we leverage LLMs to iteratively refine the initial utterances. The LLM is first tasked to curate samples that improve the dataset breadth-wise, guided by explicit diversity enhancement criteria embedded in the prompt and reinforced by strict structural constraints. Afterwards, we apply an iterative refinement process to construct a multi-tiered dataset encompassing utterances of varying complexities. In the process, we take the base utterance $U_i$, and create a deeper version $U_{i+1}$ through a specific refinement method for each category. $U_{i+1}$ can then be fed as input to the next refinement step, to get an even more challenging $U_{i+2}$ and so on. According to our experiments, the LLMs are able to generate strong refinements if we provide detailed criteria in the instruction and three refinement steps are sufficient. We share the prompts we use for all the categories in the Appendix A, and an example refinement for Paralinguistics and Foreign Words category is shown in Figures 1 and 2, respectively. Here are the description of the six categories in the final dataset:

**Questions:** Contain sequential questions and statements. This evaluates the TTS system's ability in generating interrogative and declarative prosody.

**Emotions:** Contains narrative text with long quoted dialogues of emotion intensification, followed by contrasting emotions.

**Paralinguistics:** Contains vocal interjections (Uhh, Hmmm), Onomatopoeia (Achoo, tick-tock, etc), Varied Emphasis markers (Capitalization, vowel elongation, syllable emphasis with hyphens), Pacing cues like ellipses (....) or punctuation (STOP.RIGHT.THERE), and stuttering (I-I-I d-didn't, so so-so-so-sorry).

**Foreign Words:** Covers 15 unique languages with idiomatically and prosodically rich phrases placed in between english text.

**Syntactic Complexity:** Covers complex text with garden-path sentences, deep nested clauses with centre embeddings, homographs, and other forms of syntactic complexity.

**Complex Pronunciation:** Texts with emails, phone numbers, URLs, Street Adresses, Location references, STEM equations, units and notations, Abbreviations-Both initialisms (pronounced letter by letter) and acronyms (pronounced as word), and tongue twisters.

**Dataset Statistics:** For five of the categories that we use from BASE-TTS, we curate a total of 70 seed utterances by appending 50 breadth-wise expanded sentences along with 20 curated by human, after this, we perform three iterative refinement steps, resulting in additional $70 * 3 = 210$ samples. This results in 280 samples each for these five categories. For "Complex Pronunciation", we curate 60 breadth-wise diverse samples from scratch, which are turned into 240 samples after three rounds of refinement. Subsequently, we add five complex short tongue twisters, each repeated multiple times. Based on our manual observation, TTS systems often struggle with repeated articulation-where a single slip can lead to a cascading effect. We report these findings in Section 4.2. Total sample count thus comes out to $280 * 5 + 240 + 5 = 1645$. Category-wise statistics are shown in the Appendix A.7.

### 3.2 Large-Audio-Language Model as Judge

Synthesizing speech for all $1,645$ benchmark utterances results in approximately $420$ minutes (or 7 hours) of audio per TTS system. Evaluating this volume of audio through human raters is both time-consuming and resource-intensive, with limited reproducibility, and the need for specialized linguistic knowledge. To address these limitations, we employ Large-Audio-Language Models (LALMs) as automatic judges. Our benchmark specifically targets aspects of speech synthesis-such as prosody, pausing, expressiveness, and pronunciation-that are not adequately captured by traditional metrics like

Word Error Rate (WER) or MOS-based quality estimators. Accurately assessing these dimensions requires a general-purpose, high-capacity audio understanding model.

We choose **Gemini 2.5 Pro** as our primary LALM-based judge due to its strong performance on established audio reasoning benchmarks such as MMAU [31] (See Appendix B for performance comparison of LALMs on audio understanding benchmarks). Notably, it leverages inference-time scaling [13, 21] before producing outputs, which aligns well with the complexity of our evaluation tasks.

To evaluate a candidate TTS system $T_i$, we compare it against a strong reference system $T_j$, chosen to have low WER to ensure high-fidelity synthesis of the benchmark utterances. For each evaluation instance, both systems generate speech for the same input, and the outputs are randomly assigned as $T_1$ and $T_2$ to avoid positional bias. The LALM judge is provided with the original text, the associated category label, and a structured evaluation prompt that includes the target evaluation dimension (e.g., prosody, emotion), scoring rubric, and detailed category-specific reasoning guidelines. The model is then presented with the audio from $T_1$, followed by a separation marker, and then the audio from $T_2$.

The LALM returns a structured json response containing natural language justifications for the performance of each system, a comparative analysis highlighting key differences-annotated as either subtle or significant-a scalar score in the range $[0, 3]$ for each system, and a final winner label: 0 for a tie, 1 if $T_1$ is preferred, and 2 for $T_2$. The prompt is designed to elicit chain-of-thought reasoning with time-stamp based analysis, and encourages the model to resolve borderline cases by articulating fine-grained distinctions and predict human-based preferences. The full judger prompts used for each category are shared in the Appendix C.3.

We adopt a win-rate-based metric to summarize performance. Let $W(T_i)$ denote the win-rate of system $T_i$ relative to the baseline $T_j$. This is computed as:

$$W(T_i) = \frac{\sum(\text{winner} = \text{index}_i) + 0.5 \cdot \sum(\text{winner} = 0)}{n}$$

where $\text{index}_i \in 1, 2$ corresponds to the randomized label assigned to $T_i$, and $n$ is the total number of comparisons. A score of $0.5$ reflects parity with the baseline, while deviations indicate relative superiority or inferiority.

This evaluation protocol enables robust, interpretable, and reproducible TTS comparison at scale. Unlike human raters, the LALM offers consistent judgments across multilingual and prosodically rich utterances, and its outputs include timestamp-grounded rationales that support fine-grained diagnostic analysis as evidenced by examples provided in the Appendix D. Our experiments in Section 4.6.2 show that the judge-based win-rate has high correlation with human preference as well.

# 4 Experiments

## 4.1 Setup

**Models Evaluated**  We evaluate seven open-source models: **Suno Bark (TTS)** [2], **Sesame-1B (TTS)** [35], **Zyphra/Zonos (TTS)** [5], **Tortoise-TTS (TTS)** [3], **MiniCPM (LALM)** [1], **Qwen2.5 Omni (LALM)** [46], and **Orpheus-TTS (TTS)** [9]. In addition, we benchmark four closed-source systems using their flagship models: **ElevenLabs' multilingual-v2 (TTS)**, **Deepgram's Aura-2 (TTS)**, **HumeAI's Octave (TTS)**, and OpenAI's **GPT-4o** suite, which includes both TTS and audio reasoning variants-`gpt-4o-mini-tts` (TTS), `gpt-4o-audio-preview-2024-12-17` (LALM), and `gpt-4o-mini-audio-preview-2024-12-17` (LALM). For models that are fine-tuned on specific voices, we pre-select some of these voices to show the main results. As we show later in Section 4.4, the final win-rate can be sensitive to the voice. In addition to the win-rate as described in Section 3.2, we follow standard practice by computing WER using `Whisper-v3-large` [28], and automated MOS(AutoMOS) scores are calculated using a fine-tuned `wav2vec2.0` model called `wv-mos` [7]. An expanded leaderboard with results from additional open-source and proprietary models is available on our Github repository.

**Prompting**  For pure TTS models such as Sesame1B, Orpheus-TTS, Aura-2, and Eleven Multilingual v2, we directly provide the utterance text. For other models, we compare a basic prompting setup (utterance only) with a **Strong Prompting** strategy, where the input is augmented with category-specific instructions (e.g., "be emotionally expressive" for the Emotions category). For HumeAI and

Table 1: Main results, WER↓ and Win-rate↑ over all categories with gpt-4o-mini-tts-alloy as baseline, † represents Strong Prompted models

| Model | Voice | Emotions | | Foreign Words | | Paralinguistics | | Complex Pronunciation | | Questions | | Syntactic Complexity | | Overall | | | |
|---|---|---|---|---|---|---|---|---|---|---|---|---|---|---|---|---|---|
| | | WER | Win-Rate | WER | Win-Rate | WER | Win-Rate | WER | Win-Rate | WER | Win-Rate | WER | Win-Rate | WER | Win-Rate | Parsing Fail | AutoMOS |
| gpt-4o-mini-tts (baseline) | Alloy | 0.72 | - | 13.45 | - | 20.55 | - | **29.90** | - | 0.42 | - | 1.04 | - | **10.61** | - | - | 4.23 |
| Suno Bark [2] | v2/en_speaker_6 | 4.31 | 0.00% | 26.11 | 10.89% | 33.26 | 6.60% | 55.88 | 8.36% | 3.01 | 15.00% | 6.07 | 12.50% | 20.71 | 8.90% | 0 | 3.61 |
| Sesame1B [35] | - | 17.07 | 7.32% | 45.27 | 10.35% | 49.63 | 18.92% | 80.97 | 7.40% | 2.74 | 31.78% | 4.30 | 18.88% | 32.32 | 15.96% | 4 | 3.38 |
| Zyphra/Zonos [5] | exampleaudio | 7.32 | 9.67% | 28.52 | 11.96% | 25.33 | 13.75% | 45.00 | 7.95% | 7.66 | 26.78% | 4.13 | 28.13% | 19.12 | 16.55% | 2 | 3.39 |
| Tortoise-TTS [3] | random | 13.04 | 17.92% | 29.61 | 10.00% | 64.93 | 14.28% | 51.87 | 1.59% | 10.44 | 28.28% | 6.35 | 30.82% | 28.62 | 17.67% | 1 | 3.03 |
| MiniCPM [1] | - | 12.36 | 31.83% | 33.46 | 6.42% | 58.48 | 21.50% | 82.15 | 1.84% | 5.21 | 32.50% | 3.08 | 37.50% | 31.40 | 22.36% | 4 | 3.54 |
| Qwen 2.5 Omni [46] | Chelsie | 1.22 | 41.18% | 26.98 | 11.07% | 57.48 | 17.44% | 64.07 | 3.30% | 12.77 | 49.28% | 1.66 | 36.96% | 26.58 | 27.07% | 7 | 4.09 |
| Qwen 2.5 Omni [46] | Chelsie | 2.41 | 41.60% | 26.77 | 11.42% | 58.44 | 20.25% | 49.51 | 6.12% | 0.87 | 51.78% | 3.47 | 38.57% | 23.03 | 28.77% | 1 | 4.09 |
| Orpheus TTS [9] | Tara | 1.81 | 31.78% | 22.31 | 17.5% | 40.94 | 39.82% | 41.04 | 10.61% | 1.48 | 39.64% | 1.63 | 38.92% | 17.71 | 30.12% | 0 | 3.76 |
| DeepGram Aura-2 | Thalia-en | 3.45 | 29.28% | 21.41 | 18.75% | 23.73 | 21.14% | 54.49 | 33.81% | 1.24 | 48.21% | 1.36 | 43.70% | 16.83 | 32.44% | 4 | **4.33** |
| 11Labs eleven multilingual v2 | Brian | **0.63** | 30.35% | 14.44 | 35.53% | 21.51 | 45.53% | 31.44 | 14.48% | 0.49 | 39.46% | 1.15 | 35.53% | 11.19 | 33.89% | 0 | 3.55 |
| HumeAI† | - | 0.83 | 61.60% | 21.05 | 34.64% | 19.84 | 36.91% | 37.14 | 34.28% | **0.38** | 43.21% | 0.93 | 44.64% | 12.85 | 42.73% | 1 | 4.18 |
| gpt-4o-mini-tts† | Alloy | 0.71 | 59.17% | **12.07** | 57.32% | 21.33 | 58.75% | 31.57 | **52.44%** | 0.66 | 52.67% | **0.84** | 57.14% | 10.76 | 56.32% | 2 | 4.20 |
| gpt-4o-mini-audio-preview | Alloy | 0.95 | 55.89% | 14.48 | 59.82% | **19.04** | 52.86% | 32.27 | 30.61% | 0.55 | 47.32% | 0.88 | 48.75% | 10.92 | 49.60% | 1 | 4.19 |
| gpt-4o-mini-audio-preview† | Alloy | 9.34 | 59.13% | 12.70 | 58.92% | 20.92 | 62.59% | 37.14 | 28.68% | 0.74 | 48.21% | 0.72 | 53.40% | 13.09 | 52.31% | 5 | 4.18 |
| gpt-4o-audio-preview | Alloy | 1.03 | 48.57% | 14.72 | 60.17% | 23.16 | 66.78% | 35.89 | 40.81% | 1.19 | 47.5% | 1.25 | 57.14% | 12.38 | 53.76% | 0 | 4.09 |
| gpt-4o-audio-preview† | Alloy | 0.93 | 61.64% | 13.75 | **62.5%** | 20.56 | 68.21% | 36.92 | 49.59% | 1.72 | 47.85% | 1.26 | 56.85 | 12.00 | 57.95% | 4 | 4.06 |
| gpt-4o-audio-preview† | Ballad | 1.82 | **88.84%** | 13.30 | 60.17% | 21.15 | **82.14%** | 35.32 | 40.40% | 1.38 | **56.96%** | 1.16 | **59.53%** | 11.87 | **65.17%** | 4 | 3.83 |

GPT-4o-mini-tts, these instructions are passed via optional style descriptors; for LALMs like Qwen 2.5 Omni and GPT-4o audio variants, they are included in the user message.

We calculate the win-rate of all evaluated models against gpt-4o-mini-tts(Alloy voice), judger temperature is set to 0.0. More details about hyper-parameters for the judger and evaluated models, along with the full prompting templates used to generate audios are provided in the Appendix C.

## 4.2 Benchmark Performance

**Overall Results:** Model performance, summarized in Table 1, reveals a broad spectrum of win-rates ranging from 8.90% to 65.17%. GPT-4o-Audio (Ballad voice) achieves the highest overall performance, with particularly strong results in the expressiveness-focused categories-88.84% in *Emotions* and 82.14% in *Paralinguistics*. Notably, only GPT-4o-mini-tts with strong prompting surpasses the 50% win-rate in the *Complex Pronunciation* category, suggesting targeted optimization by OpenAI for this capability. HumeAI ranks as the second-best closed-source system, outperforming Deepgram's Aura-2 (Thalia) and ElevenLabs' Multilingual v2 (Brian). The low performance of Aura-2 in multilingual settings aligns with its lack of explicit multilingual support; when the *Foreign Words* category is excluded, its win-rate rises to approximately 35%, slightly above ElevenLabs. Among open-source models, Orpheus-TTS performs best, with Qwen 2.5 Omni following closely. In contrast, Bark and Sesame1B exhibit significant performance deficits, particularly in the *Emotions* category. All open-source models perform very poorly on the *Complex Pronunciation* category. We observe that **strong prompting** consistently enhances performance for all models where both prompted and unprompted evaluations are available. For example, GPT-4o-mini-tts reaches a 56% win-rate under strong prompting, showing a clear improvement over its baseline configuration. A similar gain is observed for GPT-4o-audio-preview. Judger parsing failures stemmed from two issues: incorrect JSON formatting or reaching maximum token limits when LLMs became trapped in repetitive reasoning loops.

**Depth-wise Performance Trends:** Figure 3 illustrates how model win-rates change across increasing refinement depths for each category. Models naturally cluster into high-performing (average win-rate > 50%) and low-performing groups. Although we might expect deeper utterances to widen this performance gap-with strong models excelling and weaker ones faltering-our findings reveal more nuanced patterns. At higher complexity levels, both models may encounter difficulties, increasing the likelihood of ties. Additionally, strong models sometimes reveal systematic weaknesses when challenged by greater complexity, while lower-performing models occasionally match or exceed the baseline by avoiding specific failure modes. Nevertheless, four of our six categories exhibit clear depth-sensitive performance trends. The exceptions are *Questions* and *Syntactic Complexity*, where more subtle prosodic expectations result in less pronounced differentiation across depths.

**Win-Rate v/s AutoMOS:** Win-rates and AutoMOS scores measure different aspects of speech quality. For instance, while some models achieve high AutoMOS scores, they may have lower win-rates, and vice-versa. Across 26 models from our expanded leaderboard on Github, the Spearman correlation between win-rates and the AutoMOS scores is only $\rho_{Spearman} = 30.77\%$. This divergence is expected, as the metrics stem from fundamentally different training paradigms. AutoMOS predictors like wv-mos are trained on listening test datasets such as the Voice Conversion Challenge

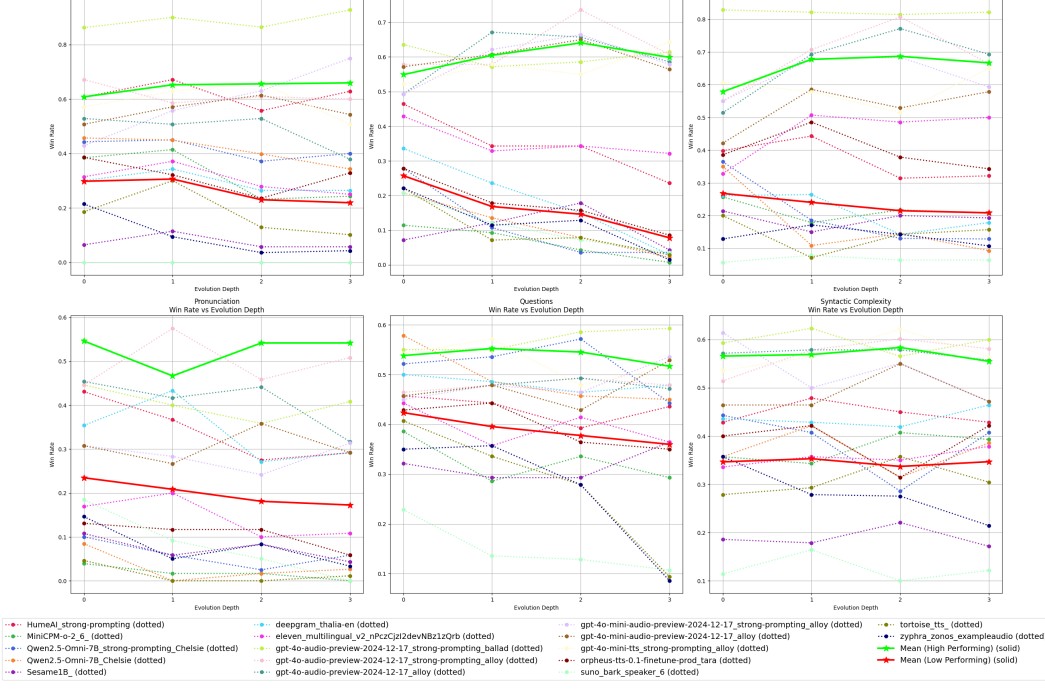

Figure 3: Win-rate chart for each category at different refinement depths. We also show the mean win-rate at each depth, computed collectively for high-performing models (average win-rate>50%) and low-performing models (average win-rate<50%).

(VCC) 2018 [22], where human raters assess general audio naturalness and signal fidelity. They are therefore optimized to detect technical artifacts but not to judge specific linguistic or expressive correctness. Our LALM-based win-rate is explicitly designed to fill this gap, a focus validated by its stronger alignment with human judgment as noted in Section 4.6.2. We therefore propose the metrics be used in complement: AutoMOS for technical quality assessment, and our win-rate for evaluating the advanced expressive and semantic capabilities of TTS models.

**Systematic Failures and Judger Insights:** Depth-wise analysis reveals consistent failure patterns and demonstrates our judger's sensitivity to prosodic, phonetic, and semantic mismatches. Most open-source models handle *Questions* and *Syntactic Complexity* adequately, with Sesame1B being the notable exception due to flat intonation and poor pausing. Sesame1B particularly struggles with *Emotions*, often inserting random interjections or producing monotonous speech. All open-source models underperform on *Complex Pronunciation*, misreading decimals, dropping digits, and breaking down at higher complexity, with MiniCPM and Tortoise-TTS failing completely even at depth 0. For *Foreign Words*, Sesame substitutes non-English tokens with unrelated content, while Orpheus anglicizes pronunciation to the extent of being phonetically incorrect.

Commercial models show different limitations: ElevenLabs falters with *Complex Pronunciation*, while Deepgram Aura-2 degrades with longer utterances and struggles with expressive *Paralinguistics*. OpenAI models excel in emotional and multilingual content but still exhibit subtle issues-occasional mispronunciations, dropped dates, and synthesis breakdowns-that our judger successfully identifies. The judger effectively distinguishes emphatic renditions, recognizes homograph disambiguation, and rewards appropriate prosody, though subtle paralinguistic nuances and emotional shifts remain challenging to evaluate perfectly. We provide a comprehensive analysis of judger behavior in Appendix D, along with quantitative evidence of the judge's audio understanding capabilities in Appendix B.

Table 2: Win-rates based on judger used, † represents Strong Prompted Models

| Judger Model → | Gemini 2.0 Flash | | Gemini 2.5 Flash | | Gpt-4o-mini-audio | | Gpt-4o-audio | | Qwen 2.5 Omni | |
|---|---|---|---|---|---|---|---|---|---|---|
| Evaluated Model ↓ | Win-Rate | Parsing Fail | Win-Rate | Parsing Fail | Win-Rate | Parsing Fail | Win-Rate | Parsing Fail | Win-Rate | Parsing Fail |
| Sesame1B | 25.30% | 3 | 24.77% | 6 | 28.60% | 2 | 31.07% | 2 | 41.39% | 76 |
| Qwen2.5 Omni Chelsie | 38.06% | 3 | 31.49% | 8 | 42.67% | 1 | 38.13% | 2 | 47.12% | 82 |
| Qwen2.5 Omni Chelsie† | 39.17% | 0 | 32.09% | 6 | 43.09% | 2 | 39.38% | 1 | 47.41% | 77 |
| Orpheus-TTS Tara | 39.41% | 1 | 38.02% | 4 | 41.03% | 0 | 41.33% | 1 | 48.59% | 74 |
| DeepGram Thalia | 40.79% | 0 | 36.27% | 2 | 43.10% | 0 | 37.26% | 0 | 47.84% | 70 |
| ElevenLabs Brian | 44.79% | 1 | 41.14% | 0 | 48.93% | 1 | 44.22% | 0 | 48.98% | 67 |
| Hume.AI | 47.99% | 1 | 40.34% | 3 | 46.20% | 0 | 47.20% | 1 | 49.42% | 76 |
| gpt-4o-mini-tts Alloy† | 54.43% | 0 | 53.43% | 0 | 52.06% | 0 | 51.51% | 1 | 50.31% | 63 |
| gpt-4o-mini-audio-preview Alloy | 48.08% | 0 | 47.14% | 0 | 48.63% | 0 | 48.72% | 1 | 50.28% | 71 |
| gpt-4o-mini-audio-preview† Alloy | 51.18% | 1 | 49.57% | 0 | 47.29% | 1 | 50.12% | 0 | 49.10% | 73 |
| gpt-4o-audio Alloy | 53.28% | 0 | 53.65% | 2 | 50.39% | 0 | 53.03% | 0 | 49.71% | 81 |
| gpt-4o-audio† Alloy | 54.98% | 1 | 57.06% | 3 | 50.54% | 0 | 54.74% | 0 | **50.69%** | 73 |
| gpt-4o-audio† Ballad | **58.78%** | 1 | **57.60%** | 1 | **55.80%** | 1 | **64.23%** | 1 | 49.30% | 68 |

## 4.3 Sensitivity to Judge

While Gemini 2.5 Pro achieves the highest performance on the MMAU [31] benchmark for audio understanding, we conducted an ablation study to assess how evaluation outcomes vary across different LALM judger models, both proprietary and open-source. Using identical audio inputs from candidate TTS systems, we varied the judger model across four closed-source and one open-source alternative. Results are shown in Table 2.

Our analysis reveals that Qwen 2.5 Omni performs poorly in the judging role, frequently producing parsing errors and yielding win-rates near 50% across the board-indicative of near-random behavior. In contrast, the remaining judger models (Gemini 2.0 Flash, Gemini 2.5 Flash, Gemini 2.5 Pro, GPT-4o-mini-audio, and GPT-4o-audio) exhibit strong agreement in their relative rankings, despite differences in absolute scores. This alignment is quantified by a high Kendall's W coefficient of concordance ($W = 0.97$), indicating near-perfect inter-model consistency and further validating the robustness of our evaluation framework.

## 4.4 Understanding bias of specific voices

Most TTS models are tied to specific voices-either through fine-tuning or voice cloning-except for a few, such as Hume.AI and Sesame1B, which generate different voice for different utterances. To examine the impact of voice identity on performance, we measure the category-wise standard deviation in win-rate across multiple voices for four models: GPT-4o-mini-tts (6 voices: Alloy, Ballad, Ash, Coral, Nova, Onyx), Deepgram Aura-2 (6 voices: Thalia-en, Andromeda-en, Helena-en, Apollo-en, Arcas-en, Aries-en), Orpheus-TTS (Tara, Leah, Jess, Leo, Dan, Mia), and Qwen 2.5 Omni (2 voices: Chelsie and Ethan). Results are shown in Figure 4a. We find that *Emotions* and *Paralinguistics* exhibit the highest sensitivity to voice variation, reflected in elevated standard deviations. This is consistent with the fact that voice fine-tuning often emphasizes expressive rendering, which these categories demand. In contrast, *Pronunciation* shows the least variance across voices, as it depends more on the inherent ability of the model and not the voice characteristics, other categories also show low variance generally.

## 4.5 Text Normalization

The main challenge of the complex pronunciation category lies in parsing uncommon characters and their groups, something that can be made easier by using Text Normalization(TN) techniques prior to sending the text to the TTS model. To this end, we do an ablation measuring the change in win-rate for various TN techniques. We also add the data point corresponding to an LLM (GPT-4.1-mini) acting as the TN, the results are in Table 3a.

We note that basic TN techniques do not always improve model performance on our benchmark and can make it worse. For instance, WeText [4] converts '$1,890.125375' to 'one thousand eight hundred and ninety point one dollars twenty five thousand three hundred and seventy five', which harms TTS quality. Similarly, '0' is sometimes normalized to the informal 'oh', which is not preferred in formal or decimal contexts. 'SQL' was correctly normalized to 'S Q L', but the baseline's pronunciation 'Sequel' was preferred. Using an LLM for TN resolves many of these issues and significantly improves win-rate, though some errors persist with the basic prompt that we used. We include more examples in the Appendix E.

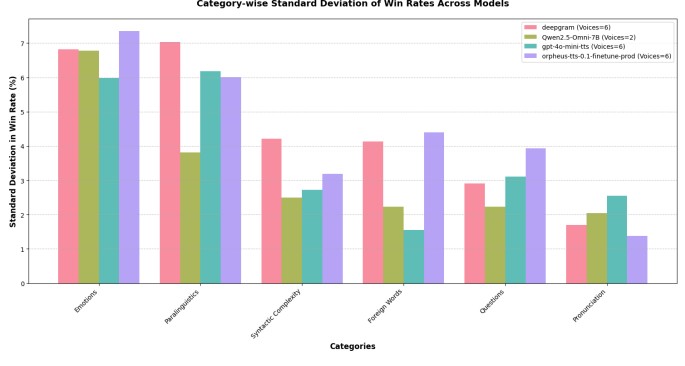

(a) Variance of win-rate by voice (gemini-2.0-flash as judge)

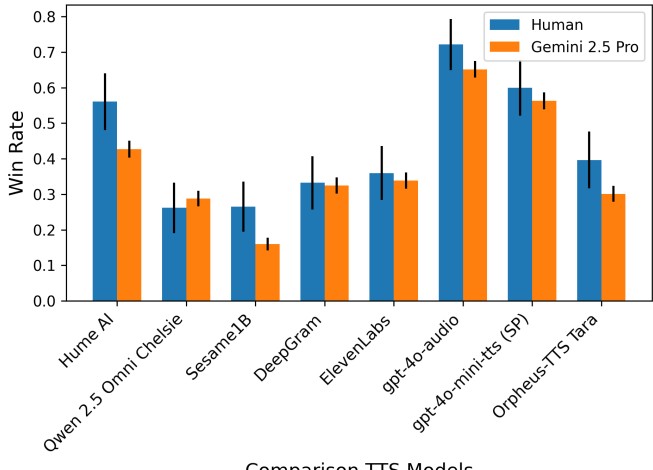

(b) Comparison of different TTS win-rates against baseline under human vs model (Gemini 2.5 Pro) with 95% CI

Figure 4: Top: Variance of win-rate by voice; Bottom: Human and model win-rate alignment.

(a) Performance difference for complex pronunciation with normalization techniques using GPT-4o-mini-TTS (Gemini-2.0-Flash judge), averaged over 6 runs.

| Text Normalization Method | Win-rate ↑ |
| --- | --- |
| No TN | 51.69% |
| WeText TN | 50.06% |
| GPT-4.1-mini TN | 76.74% |

(b) Spearman correlation between human and model judge rankings based on win-rate.

| Model Judge | $\rho_{\text{Spearman}}$ ↑ |
| --- | --- |
| Gemini 2.5 Pro | 90.5% |
| Gemini 2.0 Flash | 90.5% |
| Gemini 2.5 Flash | 90.5% |
| GPT-4o-audio | 90.5% |
| Qwen 2.5 Omni | 88.1% |
| GPT-4o-mini-audio | 76.2% |

Table 3: Left: Ablation study on the impact of different text normalization methods. Right: Correlation between human preference and different judge models.

## 4.6 Human-Model Alignment

### 4.6.1 Human Study Setup

We conducted human evaluation to measure the correlation between the model judges' preference to that of human judges. We created an online survey using Gradio, where human judges were presented with pairs of audio clips generated by a baseline TTS and a comparison TTS and instructed

to rate which is the better one (or tie). To ensure consistency in evaluation, participants were given instructions and evaluation criteria adapted from the prompts used for the model judges. The human preferences were then aggregated to compute the win-rate of each comparison model against the baseline, which was compared to the win-rates produced by model judges. For this study, we selected gpt-4o-mini-tts as the baseline and compared it against eight other models: Sesame1B, Deepgram, ElevenLabs, gpt-4o-mini-audio-preview, gpt-4o-mini-tts (SP), Hume AI, Orpheus-TTS Tara, and Qwen2.5-Omni Chelsie. These comparisons were evaluated by the following model judges: Gemini 2.5 Pro, Gemini 2.0 Flash, Gemini 2.5 Flash, GPT-4o-mini-audio, GPT-4o-audio, and Qwen 2.5 Omni.

A total of 512 audio pairs were sampled from these comparisons to ensure coverage across different categories and refinement depths. These were distributed among $N = 8$ human judges, with each judge assigned between 149 and 150 audio pairs with some redundancy among the judges.

### 4.6.2 Agreement Between Human and Model Judgements

To evaluate alignment between human and model judgments, we computed the Spearman correlation between the comparison models' rankings based on win-rates derived from human ratings and those derived from each model judge. As shown in Table 3b, all judges achieved high correlation scores of upto 90.5%, suggesting that model judges closely mirrors human preferences in determining which TTS system performs better. AutoMOS scores achieve a low correlation $\rho_{Spearman} = 21.43\%$ with human preference, underscoring the necessity of our LALM-based win-rate to complement traditional metrics for a more complete evaluation. We also analyzed the individual win rates of each comparison model (vs. the baseline) under both human and model evaluations. As shown in Figure 4b, most model win rates are closely aligned with human judgment (within 95% CI), though discrepancies exist in some cases (e.g., Hume AI, Sesame1B), where the model (Gemini 2.5 Pro) over-estimates performance compared to human preference.

## 5 Limitations and Conclusion

There are two main limitations to our work related to the dataset creation and the LALM-as-judge paradigm. First, LALMs have inherent biases that may manifest in our synthetic dataset, such as preferences for literary language and formal phrasing patterns. For categories like Foreign Words and Syntactic Complexity, refinement level of depth=3 produces grammatically correct but somewhat artificial utterances that occur infrequently in natural communication, but still act as a solid stress-test for TTS systems. Additionally, our multilingual evaluation focuses on Latin transcriptions rather than native character sets, which doesn't fully capture the challenges of true multilingual TTS. Regarding evaluation, using Gemini 2.5 Pro incurs substantial costs-approximately $50 per complete TTS system evaluation. Nevertheless, the strong ranking agreement observed across different judge models suggests opportunities for more economical alternatives without significant quality loss. We also observe that evaluating subjective aspects like emotions, prosody, and intonation can occasionally lead to LALM hallucinations, where judges incorrectly identify pronunciation issues. Despite these considerations, EmergentTTS-Eval represents a significant advancement in TTS evaluation methodology by addressing critical gaps in existing benchmarks. Our approach systematically challenges TTS systems across dimensions that conventional metrics overlook, while offering a scalable alternative to resource-intensive human evaluations. The strong correlation between our LALM judges and human preferences validates the approach, while the benchmark's ability to reveal fine-grained performance differences demonstrates its practical utility for driving progress in creating more human-like synthetic speech.

## 6 Broader Impacts

EmergentTTS-Eval aims to accelerate the development of more expressive, accurate, and inclusive TTS systems, which can greatly benefit accessibility tools and enable more natural conversational interfaces across a variety of applications. However, highly convincing TTS systems could be used to perpetrate fraud or spread disinformation, and LALM judges may perpetuate biases. To mitigate these risks, we encourage pairing TTS systems with deepfake detectors or watermarking and auditing prompt and judge outputs for diverse representation.

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

# APPENDIX

The appendix is organized as follows:
**Section A**: Prompts and details for breadth and depth refinement of each category, along with final dataset statistics.
**Section B**: Benchmarking of LALMs on audio understanding tasks.
**Section C**: Evaluation related details, such as hyperparameters, audio generation prompts, and prompts for the judger.
**Section D**: Analysis of Gemini-2.5-Pro as a judger and the case of Audio Subjectivity.
**Section E**: Text Normalization prompt and examples.

## A   Per-Category Depth and Breadth Refinement

For breadth-expansion, we leverage long-thinking LLMs like Gemini 2.5 Pro, GPT-o3 and Claude 3.7 Sonnet. We prompt all three with the same breadth prompt, and through manual analysis select the LLM which produces the best breadth expansion and report the same for each category below. Next, all depth-refinements are achieved using Gemini-2.5-Pro. We create the following template, which is populated with the `**text_to_synthesize**`, and the refinement method for each category `**complex_prompt_method**`. The depth refinement prompt has a field `tts_synthesis_diversity` required in the LLM output, this is the field where COT specifications are provided for each category to ensure high-quality diverse and complex refinements from the LLM. The template is:

```
"""
You will you act as a Text Rewriter for a piece of text **text_to_synthesize**,
    which is the text corresponding to which a TTS system has to synthesize
    realistic speech.
Your objective is to rewrite and evolve the given text into a more complex
    version **rewritten_text_to_synthesize**, which makes the famous Speech
    Generation AI systems (e.g., ElevenLabs, Deepgram) harder to handle as
    compared to the original **text_to_synthesize**.
The underlying goal is to be come up with the **rewritten_text_to_synthesize**
    such that,
**It is more complex, deep and harder for a TTS system to synthesize than **
    text_to_synthesize**.**
You WILL complicate and complexify the given **text_to_synthesize** using the
    following method:
**complex_prompt_method**
{{{complex_prompt_method}}}
**/complex_prompt_method**
Now, you will be provided with the **text_to_synthesize**
**text_to_synthesize**
{{{text_to_synthesize}}}
**/text_to_synthesize**

**Output Format:**
You will output a json object with the following fields:
```json
{
    "text_to_synthesize": str <Verbatim copy of the original text to synthesize>,

    "tts_synthesis_diversity": str <Reasoning on how you can complicate the **
        text_to_synthesize** using the details specified in **
        complex_prompt_method** to make it more challenging for the TTS system
        to synthesize>,
    "rewritten_text_to_synthesize": str <The rewritten text the TTS system has
        to synthesize which is more complex and deep than **text_to_synthesize
        **>
}
**/Output Format:**
Now, you will output the correct json object following the **Output Format:**
    and without producing any additional text.
```

```
        """
```

In the following sections, we present the category-wise breadth expansion prompts and the `complex_prompt_method` used for each category.

## A.1 Category 1: Questions

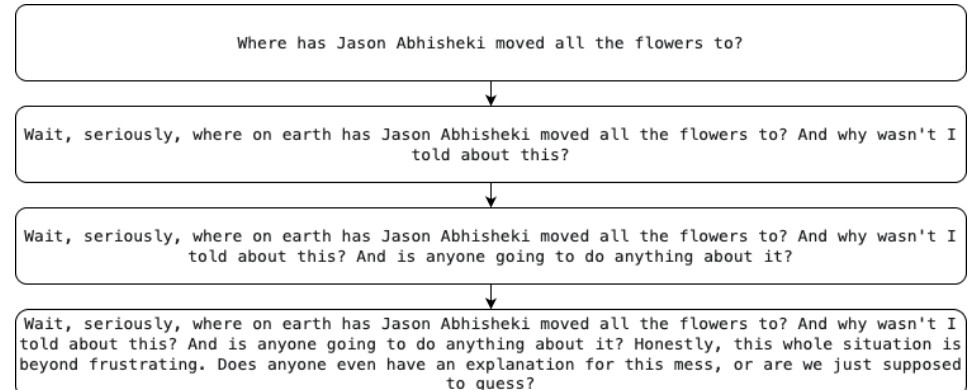

Figure 5: Example depth-refinement for questions category. Starting with a simple Wh-question, complexity is introduced by first by adding a subsequent Wh-question with a pragmatic nuance, then a Yes/No question to test pitch contour shifts. The final refinement examines the differentiation between interrogative and declarative prosody by inserting an emphatic statement, and further tests nuanced intonation with a concluding alternative question

**Breadth Expansion**   We use 20 samples that were proposed for this category in BASE-TTS, and then prompt **Gemini-2.5-Pro** to curate 50 more samples that achieve a significantly wider coverage for evaluating TTS interrogative prosody. Beyond standard Yes/No and Wh- questions, we incorporate negative questions, rhetorical questions, declarative questions expressing surprise or doubt, hypotheticals, alternative questions involving lists, and questions featuring parenthetical elements or starting conjunctions, to get a total of 70 samples with richer syntactic diversity and broader prosodic demands. The breadth expansion prompt used is:

```
Consider the below set of 20 samples. This set belongs to the "Questions" category
    and is used for create an extremely diverse set for evaluation of TTS systems,
    where they have to synthesize the text corresponding to **text_to_synthesize**.
This category evaluates whether the system correctly applies question intonation
    patterns. Questions usually have a distinct pitch movement, often rising at the
     end in yes/no questions, while wh-questions may have a more neutral or falling
     tone, and there are many other scenarios that can be used to evaluate a TTS
    system not covered in the 20 samples.
Your goal is to generate 50 more samples belonging to this category. You will do
    this in the following step-by-step manner:
1. You will analyze the 20 samples carefully.
1.a. Reason deeply about the types of questions this set contains sample-by-sample,
    and the corresponding intonation patterns that these questions might elicit
    from our TTS system. Remember that a single text can have variability in
    intonation pattern, but we have to form an abstract map of the patterns that we
     are seeing.
1.b Reason deeply about the **text_to_synthesize** structures present, like the
    placement of question marks, number of questions marks, the grammatical
    structure of the texts.
2. Now, you will think long about what this set is **MISSING**, specifically, the
    various types of questions that exist in the complete set of english texts, but
     are not present in this set, **AND** will be great to test the intonation
    pattern of TTS systems on.
3. Finally, you will create additional 50 samples, that expand the current 20 sample
     set in terms of **DIVERSITY**, as you are doing a breadth-wise evolution of
    this 20 set.
```

```
The main goals for the set are:
1. There should be diversity in elicited intonation patterns and types of questions.
2. There should be diversity in terms of sentence grammatical structure.
3. No sentence should contain cues like * or **.
4. The 50 samples will follow the same format as the 20 samples, **BUT** no sample
    in the 50 set should be similar to what is in the 20 samples, in terms of
    context and phrasing.
5. You will not add diversity through italics, bold, UPPERCASE, etc.

Now, you are given the 20 sample set, after this, think deeply and create the 50
    question set.
'''jsonl
<now the 20 samples were provided in jsonl format>
'''
```

**Depth Refinement**   To move beyond simple questions, we utilize the depth refinement strategy to generate utterances demanding highly varied interrogative and declarative prosody from the TTS system. We refine iteratively using one of three methods: appending a sequential question, appending a statement and then a question, or infusing pragmatic nuance before appending a question. The resulting dataset is designed to specifically evaluate a TTS system's proficiency in: (a) rendering natural pitch contours across consecutive questions, (b) executing smooth prosodic transitions between declarative and interrogative speech within one utterance, and (c) conveying subtle pragmatic meanings (like skepticism or politeness) through appropriate intonational variation alongside the core question structure. Refer to Figure 5 for an example refinement and the depth-refinement prompt is as follows:

```
complex_prompt_method="""
    You are evolving a **text_to_synthesize** sample belonging to the "Questions"
        category for evaluating Text-to-Speech (TTS) systems.
The primary goal of this category is to test if the TTS system correctly applies
    natural, varied, and appropriate **question intonation patterns** reflecting
    different question types, pragmatic functions, and attitudes, especially in
    sequence.

Your task is to **increase the prosodic complexity related specifically to the act
    of questioning and its conversational context**, making the task more
    challenging for the TTS system's intonation capabilities. Apply **ONE** of the
    following evolution methods:

**Choose ONE Method per Evolution:**

1. **Method 1: Add Sequential Question:** Add *one* related, grammatically complete
    question *immediately following* the existing text. The goal is to create a
    sequence that tests the TTS system's ability to naturally transition
    prosodically between two consecutive questions, potentially involving different
     question types or nuances **to test varied intonation patterns**.

2. **Method 2: Add Sequential Statement + Question:** Add *one* related,
    grammatically complete statement *immediately following* the existing text, and
     then add *one* related, grammatically complete question *immediately following
     that statement*. The goal is to test the TTS system's ability to naturally
    transition prosodically from the original text's context, through a declarative
     statement (with appropriate intonation), and into a final question (with
    appropriate interrogative intonation).

3. **Method 3: Infuse Pragmatic Nuance & Add Sequential Question:** First, **modify
    the existing text** by adding or changing words/phrases (e.g., adverbs,
    introductory phrases, discourse markers, slight rephrasing) to require the TTS
    system to convey a specific **attitude or pragmatic function** (like doubt,
    surprise, politeness, insistence, etc.) primarily through prosody. Then, **add
    one** related, grammatically complete question *immediately following* the
    modified text. The goal is to test the TTS system's ability both to render the
```

```
        subtle nuance in the first part and to transition naturally into the
        appropriate intonation for the subsequent question.

**Crucial Constraints:**

* The final evolved text must remain a **natural-sounding, self-contained**
        utterance spoken by a **single speaker**.
* The modification should primarily challenge the **prosodic rendering** (intonation
        , pitch, stress, rhythm, phrasing, pauses) related to the **questioning
        function, attitude, or sequence**.
* **IMPORTANT:** Avoid significantly increasing **internal grammatical complexity**
        (e.g., complex clauses, deep nesting, excessive parentheticals) *unless* it is
        a direct and necessary result of naturally expressing the pragmatic nuance or
        creating the statement/question sequence described in the methods. The goal is
        **question prosody diversity**, not primarily syntactic parsing difficulty.
* **IMPORTANT:** Do **NOT** use formatting characters like bold ('**'), italics
        ('*'), or ALL CAPS to indicate emphasis or complexity. The challenge must come
        from the text structure and implied prosody itself.

In the 'tts_synthesis_diversity' field:
    1. Clearly state **which Method (1, 2, or 3)** you applied.
    2. Explain **specifically how** this modification increases the challenge for a
        TTS system's **question prosody rendering**, focusing on the required
        intonation patterns, pitch movements, stress placement, phrasing, timing, or
        the need to convey subtle nuances and manage transitions compared to the
        input text. Avoid focusing justification solely on syntactic structure.
"""
```

## A.2   Category 2: Foreign Words

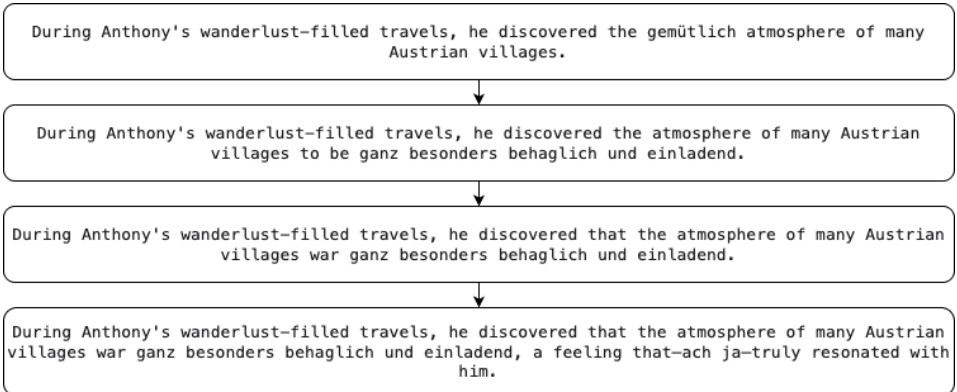

Figure 6: Example depth-refinement for foreign words category. Starting with a text containing one isolated foreign words, we expand it with german phrase. Then, the english word "to be" is replace with "war". In the final evolution, new text is added in the suffix with english and german words.

**Breadth Expansion**    The initial set of 20 BASE-TTS samples showed several constraints - they primarily featured European languages and frequently used easier to pronounce loanwords as the foreign word. To improve variety, we employed **GPT-o3** to create an additional 50 samples. This expansion significantly enhances language diversity by incorporating more samples from the 10 most spoken languages around the world (Mandarin Chinese, Hindi, Spanish, French, Arabic, Russian, Portuguese, Japanese, German and Indonesian), with particular emphasis on uncommon foreign vocabulary that presents pronunciation challenges. All non-English words appear exclusively in standard Latin characters without using any special foreign alphabetic symbols; this design choice is intentional and follows our reasoning for testing the emergent capabilities of TTS systems, and while multilingual training data with foreign character set might be highly available, the same is very limited or non-existent with Latin characters only. The breadth expansion prompt used is as follows:

Consider the below set of 20 samples. This set belongs to the "Foreign Words" category and you will use it to create an extremely diverse set for evaluation of TTS systems, where the systems have to synthesize the text corresponding to **text_to_synthesize**.

This category tests whether the system correctly pronounces foreign words and phrases, either using their original pronunciation or a widely accepted anglicized version, and there are many other foreign words that can be used to evaluate a TTS system which may not be covered in the 20 samples.

Your goal is to generate 50 more samples belonging to this category. You will do this in the following step-by-step manner:

1. You will analyze the 20 samples carefully.
1.a. Reason deeply about the types of languages covered and the commonness of the foreign words among bi-lingual population that speak english and the language of that foreign word belongs to in the sample.
1.b Reason deeply about the **text_to_synthesize** structures present, like the placement of foreign words, the number of foreign words, and the grammatical structure of the texts.
2. Now, you will think long about what this set is **MISSING**, specifically, the various types of foreign words that exist in the **COMPLETE** set of foreign words used by bi-llingual speakers, but are not present in this set, **AND** will be great to test foreign word synthesis ability of TTS systems.
3. Finally, you will create additional 50 samples, that expand the current 20 sample set in terms of **DIVERSITY**, as you are doing a breadth-wise evolution of this 20 set.

The main goals for the set are:
1. The additional 50 samples will come from these 10 wide-spread languages(5 from each language): Mandarin Chinese, Hindi, Spanish, French, Modern Arabic, Russian, Portuguese, Japanese, German and Indonesian (Malay). Ensure that each language has atleast 1 foreign phrase
2. **All text_to_synthesize that will be generated in the 50 samples, should form a fluent sentence that a bi-lingual speaker incorporating that language might speak. This means, we don't have to write the english translation or synonym of that word along with the foreign word, its **one single fluent** sentence without redundancy and completely natural flow.**
3. There should be diversity in terms of placement position of foreign words, and the number of foreign words should either be 2 or 3 across samples.
4. All foreign words must be places inside "quotes", and should naturally intergrate with the surrounding context.
5. You may incorporate small foreign phrases like the initial 20-sample set has joie de vivre, pick carefully where you want phrases, and **DO NOT** include the phrases in the quotes "", only words are to be included in quotes.
6. The 50 samples will follow the same JSONL format as the 20 samples, **BUT** no sample in the 50 set should have the same foreign word as the foreign words in the 20 sample set. All foreign words **MUST** not be duplicated.
7. You will not create texts with * word * or ** word ** or add text inside parenthesis () to indicate something.
8. **AVOID LOANWORDS: All the foreign words generated in the 50 sample set should be uncommon and slightly complex to pronounce correctly in that language, we don't want to include words that may have an easy pronunciation for an english speaker. AT THE SAME TIME, do not use very obscure words that don't exist in the modern vocabulary, we want to create natural day-to-day sentences that bi-lingual speakers will speak**
9. **You will NOT adopt characters from foreign languages, all foreign words must be expressed with english and latin letters.**
10. In the "misc" field, add a new key "foreign_language", and populate it with the language of that sample

Now, you are given the 20 sample set, after this, think deeply and create the 50 foreign words set.
```jsonl
<now the 20 samples were provided in jsonl format>
```

**Depth Refinement**   While evaluating TTS on isolated foreign words tests basic pronunciation, it doesn't reflect the full complexity of natural bilingual communication, which frequently integrates longer foreign phrases, we fill this gap through the depth refinement strategy. This method systematically transforms simpler utterances into variants featuring more substantial foreign language segments, mimicking how bilingual individuals speak and write. Guided by a specific prompt, an LLM applies one of three approaches: (i) replacing isolated words with idiomatic phrases, (ii) expanding short phrases by absorbing and translating adjacent English context, or (iii) appending bilingual affixes to already complex sentences. This yields utterances requiring the TTS system to manage fluent code-switching and natural prosody over extended foreign elements, providing a more rigorous assessment of its capabilities. This refinement strategy resulted in complex code-switching, but due to grammatical differences between languages, often created awkward sentences by the final refinement. To remedy this, we post-process the output of each refinement step through a separate LLM call to gemini-2.5-pro, to fix any grammatical and syntactical issues, we found this to be quite effective. Figure 6 illustrates this process, and the depth-refinement prompt is as follows:

```
    complex_prompt_method = """
The **text_to_synthesize** belongs to the "Foreign Words" category.
This category evaluates weather the system can fluently pronounce foreign words and
    phrases, smoothly switching between different languages within one **
    text_to_synthesize**.
Your goal: Increase TTS synthesis complexity by adding exactly **one natural, fluent
    bilingual flourish**.

**EVOLUTION LOGIC (you will choose ONE of the following approaches):**

Approach 1: Expand an **ISOLATED** Foreign Word
- **Condition**:
    - For this approach to apply, the **text_to_synthesize** will contain ATLEAST
        ONE **ISOLATED** foreign word.
    - Additional information:
        - By **ISOLATED**, we mean it's a single individual word or a compound noun
            (like 'jamon iberico' or 'Feng Shui') that functions as one unit.
        - A word is not isolated even if it is a single unit if it is surrouneded by
            foreign words. It should be <english_word> <isolated_word_or_unit> <
            english_word> for this approach to apply.
        - These **ISOLATED** words can be within "quotes" or be unquoted.
        - The **text_to_synthesize** may contain multiple such **ISOLATED** words.
- **Transformation**
    - Choose one of the **ISOLATED** foreign word, and replace it with a one of the
        following:
        - A longer prosodically rich phrase.
        - A longer idiomatic phrase(ONLY IF IT FITS THE CONTEXT EXTREMELY WELL).
        - This phrase should **MOSTLY** be in the foreign language, but can have a
            few english words to make it natural and test rapid code-switching.
        - This phrase can be another way of saying the **ISOLATED** word, **OR** is
            a replacement for the **ISOLATED** word while maintaining sense of the
            surrounding text.
    - Weather or not the the **ISOLATED** foreign word is in quotes, the phrase you
        will replace it with will not be within quotes(unless it is a dialogue).
    - Choose the **ISOLATED** word that will support the most natural expansion from
        word to idiomatic or prosodically rich phrase.
    - The added phrase should be **ATMOST** 5 words long.

**If the condition for Approach 1 is not satisfied, you will move to Approach 2**

Approach 2: **Grow** an Existing **SHORT** Foreign Phrase by **Absorbing** English
    Context
- **Condition**:
    - The **text_to_synthesize** DOES NOT contain any **ISOLATED** foreign words.
    - The **text_to_synthesize** contains **foreign phrases** where *ATLEAST ONE*
        phrase is no more than **5 words long**. (See Global Constraints for '
        foreign phrase' definition).
    - The phrases that follow the condition are called **SHORT FOREIGN PHRASE**.
- **Transformation**:
```

- **Select** one **SHORT FOREIGN PHRASE** (<= 5 words).
- **Identify adjacent English words** (immediately before and/or after the selected phrase) that can be naturally incorporated into the foreign language segment.
- **You will identify atleast **TWO** adjacent English words** that can be naturally incorporated into the foreign language segment.
- **Convert** these adjacent English words into the **same foreign language** as the phrase.
- **Integrate** the original short foreign phrase and the newly translated words into a single, **longer continuous segment** of the foreign language.
- **Crucially**: Make necessary **grammatical adjustments** *within this expanded foreign segment* for fluency and correctness in the foreign language (e.g., adjust word order, add/remove articles/prepositions, use correct verb conjugations or noun declensions as needed in that language).
- **Ensure** the transition from English into this longer foreign segment, and back into English afterwards, remains smooth and the overall sentence is fluent and grammatically sound.
- The goal is to create a **longer continuous block** of the foreign language, testing sustained synthesis and integration.
- Choose the short phrase and surrounding English context that allows for the most natural grammatical integration and fluent expansion into a longer foreign segment.
- **Do NOT apply** this transformation to any foreign phrase that is already ** more than 5 words** long in the original text.
- If all foreign phrases are already long (> 5 words), skip to Approach 3.

**If both the conditions for Approach 1 and Approach 2 are not satisfied, you will move to Approach 3**

Approach 3: Insert additional text with a **NEW FOREIGN PHRASE** (Prefix or Suffix)
- **Condition**:
  - If the conditions for Approach 1 and Approach 2 are not satisfied, you will move to Approach 3.
- **Transformation**:
  - Add additional text with english words **AND** a **NEW FOREIGN PHRASE** in the same foreign language already used in the utterance that is idiomatic and prosodically rich.
  - The new foreign phrase will be a either a **prefix** or **suffix** to the ** text_to_synthesize**.
  - Choose one of:
    **Prefix**: Add at the start of the sentence, as a lead-in.
    **Suffix**: Add at the end, as a reflective or emotional continuation.
  - Inserted phrase must be **plausible, fluent, narratable by a bilingual speaker ** and **MUST** contain words borrowed from both english and the foreign language.
  - The newly added text must be **ATMOST** 10 words long, including the english words and the foreign phrase.
---

**Global Constraints (Always Apply):**
- Use **only English + one foreign language** (same throughout the utterance).
- There will always be some foreign word or phrase in the text that you have to recognize correctly.
- All foreign words/phrases must be in **Latin transcription** AND pinyin transcription for Chinese. (no native scripts or characters from the foreign language).
- **Definition**: A 'foreign phrase' specifically refers to **a contiguous sequence of two or more words** from the foreign language, with no English words interrupting that sequence. Single words or compound nouns acting as single units are considered 'isolated words' for Approach 1.
- You **WILL** do paraphrasing of the text after the editing to make it more natural and eliminate any redundant text/unnatural text.
- Use commas, dashes, or natural punctuation to integrate long prefixes and suffixes and foreign phrases.

```
- Do not use quotation marks around foreign phrases. But if quotes are used to
    represent dialogues, you will preserve them. Only quotes that signify that a
    word is a foreign word will be removed.
---

**Phonetic & Prosodic Complexity (TTS Focus)**
- Expanded or inserted phrases should enhance TTS difficulty via:
    - Nasal vowels
    - Consonant clusters
    - Rolling R's, vowel alternation, liaison
    - Multisyllabic cadence, rhythmic variation
- Never sacrifice fluency, narratability, or naturalness.

---

In your 'tts_synthesis_diversity' explanation, clearly and consistently state:
1. What approach (1, 2, or 3) was applied and **why**.
2. What foreign language is present in the **text_to_synthesize** other than English
    .
3. What specific word(Approach 1) or phrase(Approach 2) will be expanded/absorbed or
     in case of Approach 3, what will be the newly added text and the foreign
    phrase. Mention how this **DOES NOT** make the text syntactically awkward.
4. The exact word count of the newly foreign phrase, and in case of Approach 3, the
     word count of the newly added text and the foreign phrase. **For Approach 2,
    state the original short phrase, the absorbed English words, and the resulting
    longer foreign segment with its word count.**
5. Phonetic/prosodic challenges introduced (e.g., nasal vowels, clusters, rhythm).
    **For Approach 2, emphasize the challenge of sustained foreign language
    synthesis and grammatical integration.**
6. **Intermediate Checking:** Based on the above details, print exactly what the
    candidate output **rewritten_text_to_synthesize** will be like.
7. Carefully analyze the current candidate **rewritten_text_to_synthesize** and
   7.1 Add any necessary punctuations in the candidate to ensure the text is
        coherent and logically correct, both **INSIDE** and **OUTSIDE** the
        idiomatic/prosodically rich foreign phrases.
   7.2 Restructure, paraphrase and edit the text grammatically to ensure **NO
        AWKWARDNESS** in the text. Analyze the gender, the noun-adjective agreement,
         the verb-subject agreement, tenses, etc.
8. Now, after all above steps, confirm that the final result:
    - Does not include **more than one foreign language**
    - Does not include **foreign phrases >5 words**, unless it is a Approach 3
        prefix/suffix **or the result of Approach 1/Approach 2 expansion/absorption
        **.
    - Does not include **foreign words/phrases** that are not in **Latin
        transcription** (no native scripts or characters from the foreign language
        are allowed).
    - The final result is a realistic text that can be narrated by a bilingual
        speaker in a day-to-day conversation or during narration of a story.
    - The final result is a **grammatically correct** and **syntactically fluent**
        text. It has **NO AWKWARDNESS** with respect to the gender and flow of the
        text.
"""
```

We also share the system message used for the post-processing step of fixing grammatical awkwardness using LLM, the text_to_synthesize is provided as the user-message.

```
post_process_prompt = """
You are given a sentence that has code-switching at muliple points between two
    languages.
Your goal is to refine the sentence, to remove any grammatical issues and
    awkwardness that is present.
- You will particularly be careful about the gender, the noun-adjective agreement,
    the verb-subject agreement, tenses, punctuations, etc.
- You will recognize if the text is syntactically incorrect or overly complex, in
    which case you will untangle it and make it syntactically easier to read.
```

```
- You will not add any characters from the foreign language, we will only use latin
    transcription and pinyin transcription for chinese.
Your goal is to return the refined sentence that is **SIMILAR** to the original
    sentence, but has **no** grammatical issues, awkwardness, unnaturalness, and is
     easier to read for a bi-lingual speaker proficient in the foreign language.
- You **WILL NOT** add markers that are meant to help with identification of the
    foreign segments, like do not add markers like * or ** that are not present in
    the original sentence.
You will output **ONLY** the refined sentence, with no other information or text.
"""
```

## A.3 Category 3: Paralinguistics

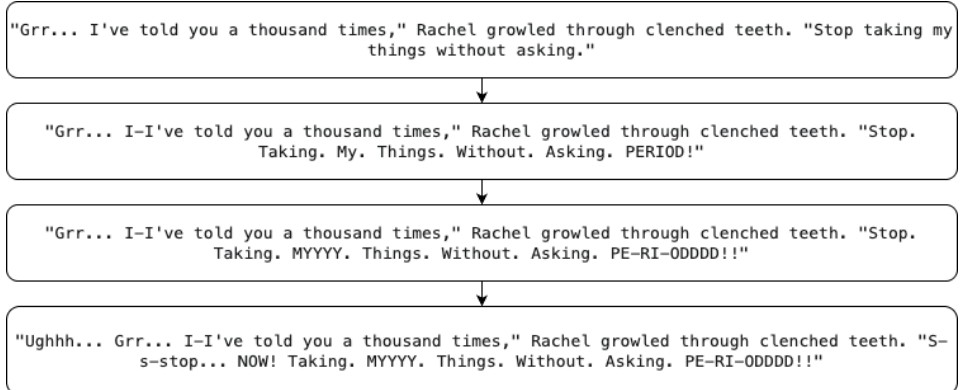

Figure 7: Example depth-refinement for paralinguisitcs category. Starting with just one cue "Grr", we add stuttering "I-I've", punctuation(Stop.Taking.My..) and caps(PERIOD). Then, syllable stress and elongation is introduced through PE-RI-ODDDD. Finally, we add more cues like "Ughhh", "s-stop" and "NOW!".

**Breadth Expansion** The initial set of 20 samples from BASE-TTS provided foundational coverage of common paralinguistic phenomena, including basic interjections (e.g., 'Aha!', 'Oops'), simple vocal sounds ('Yawn', 'psst'), and common hesitations ('Uh', 'Hmm'). However, this initial set lacked significant diversity. It underrepresented a wider spectrum of emotional interjections, varied onomatopoeia (spanning bodily, environmental, and animal sounds), nuanced textual emphasis cues, explicit pacing markers, and complex speech disfluencies. The breadth-expanded set with 50 additional samples significantly addressed these gaps by incorporating 5 types of cues such as: (i) additional interjections (e.g., Eww, Gasp, Tsk tsk, Oy vey), (ii) diverse onomatopoeia (e.g., Achoo, Tick-tock, Pitter-patter), (iii) varied emphasis markers (e.g., capitalization like REALLY, vowel elongation like sooooo, hyphenation like ab-so-lutely or Un-der-STAND), (iv) pacing cues via ellipses (...) or punctuation (STOP. RIGHT. THERE.), and (v) stuttering representations (e.g., I-I, N-N-NO). The breadth-expansion is achieved using **Claude 3.7 Sonnet** using the following prompt:

```
Consider the below set of 20 samples. This set belongs to the "Paralinguistics"
    category and you will use it to create an extremely diverse set for evaluation
    of TTS systems, where the systems have to synthesize the text corresponding to
    **text_to_synthesize**.
This category evaluates how well the system interprets **textual cues**-like
    interjections ("Wow!"), vocal sounds/onomatopoeia/animal sounds ("Shhh!", "
    Achoo!"), emphasis (CAPS, "sooo",ab-SO-lutely), punctuations ("?!"), or
    hesitations ("Uh...")-to produce appropriate **non-neutral vocal effects**, and
     there are many other paralinguistic cues that can be used to evaluate a TTS
    system which may not covered in the 20 samples.
Your goal is to generate 50 more samples belonging to this category. You will do
    this in the following step-by-step manner:
1. You will analyze the 20 samples carefully.
```

```
1.a. Reason deeply about the types of paralinguistic cue this set contains sample-by
     -sample, and the corresponding sounds that these questions might elicit from
     our TTS system.
1.b Reason deeply about the **text_to_synthesize** structures present, like the
     placement of paralinguistic cues, and the grammatical structure of the texts.
2. Now, you will think long about what this set is **MISSING**, specifically, the
     various types of paralinguistic cues that exist in the **COMPLETE** set of
     paralinguistic cues present in english texts, but are not present in this set,
     **AND** will be great to test paralinguistic speech synthesis ability of TTS
     systems. You *WILL* think about abstract types of paralinguistic cues, and then
      expand on what kind of cues they contain, for example, "Uh", "Uhmm", etc, all
     belong to the sample abstract type.
3. Finally, you will create additional 50 samples, that expand the current 20 sample
      set in terms of **DIVERSITY**, as you are doing a breadth-wise evolution of
     this 20 set.

The main goals for the set are:
1. There should be diversity in types of paralinguistic cues present.
2. **All the cues present in the 50 set you generate, should be realistically
     synthesized by a human. These should be sounds that humans can produce well
     based on the textual cue.**
3. There should be diversity in terms of sentence grammatical structure.
4. The 50 samples will follow the same JSONL format as the 20 samples, **BUT** no
     sample in the 50 set should be similar to what is in the 20 samples, in terms
     of context and phrasing and paralinguistic cue present.
5. You will not create texts with * word * or ** word ** or add text inside
     parenthesis () to indicate something, the TTS system will synthesize cues
     directly present in the text itself.

Now, you are given the 20 sample set, after this, think deeply and create the 50
     paralinguistic set.

‘‘‘jsonl
<now the 20 samples were provided in jsonl format>
‘‘‘
```

**Depth Refinement**    To get a representative set of paralinguistic cues that occur in written dialogue aiming to convey expressive speech, such as found in scripts, fictional narratives, and certain forms of informal communication, we apply the depth-refinement strategy. The refinement prompt uses the 5 defined types of paralinguistic cues and rewrites the text by incrementally adding one more or two cues of any type at each step. By the final refinement step, this process yields texts with multiple distinct paralinguistic cues, designed to work together to create a unique and realistic challenge for the TTS system. An example refinement is shown in Figure 7, and the prompt used is:

```
complex_prompt_method = """
You are tasked with enhancing the paralinguistic complexity of the provided **
     text_to_synthesize**, which belongs to the "Paralinguistics" category. This
     category tests a TTS system's ability to render non-lexical vocal cues (emotion,
      emphasis, sounds, hesitations, etc.) *within the text itself*, using
     appropriate prosody, pacing, and vocal quality.

Your specific task is to rewrite the **text_to_synthesize** by ADDING ONE or TWO
     more complex paralinguistic cues. Perform the following steps:

1. **Analyze the existing paralinguistic cues** within the spoken text and identify
     opportunities for enhancement.
2. **Enhance paralinguistic complexity using ONE appropriate technique that modifies
      the *spoken text itself*.** Choose a method that fits naturally and logically
     within the context. Examples of techniques include, but are not limited to:
   * Adding **Interjections or Filled Pauses** directly into the speech (e.g., "Wow
       !", "Gosh", "Hmm", "Uh...", "Mmm,").
   * Adding **Onomatopoeia or Explicit Vocal Sounds** as part of the utterance (e.g
       ., "Crash!", "Achoo!", "Shhh", "Psst").
```

* Introducing/Intensifying **Emphasis Cues *within the text*** (e.g., using ALL CAPS for stressedwords, using **hyphenation/syllable stress like 'im-por-tant'**, **repeating letters like 'heyyyyyy'**, using expressive punctuation like '......' or sequences like '. . .').
* Introducing **Hesitation/Stuttering Cues** directly in the speech (e.g., 'I-I-I', 'W-we-well..., ok-ok-ok').
* Modifying phrasing or punctuation *within the speech* to suggest specific **Pacing/Rhythm** (e.g., using **ellipses '...'**, short staccato phrases connected by hyphens or commas).
** NOTE: The above provided examples for each technique are only for your reference, you may add them if it fits the context, but come up with your paralinguistic cues that fit the technique and the context.**
3. **AVOID techniques relying on meta-textual instructions or non-standard formatting:** Do **NOT** add parenthetical descriptions like '(laughing)', '(angrily)', descriptive dialogue tags like '*he whined*', or **any markdown formatting like '**word**' or '*word*'** to indicate how the text should be spoken. Focus only on cues the TTS would encounter in the direct text-to-be-synthesized.
4. **Complexity Goal:** The technique should aim to *increase the demand* **SIGNIFICANTLY** on the TTS system to produce specific, non-neutral vocal delivery, without sacrificing realism or clarity.
5. **CRITICAL CONSTRAINT - Realism & Clarity:**
    * Your **absolute priority** is ensuring the rewritten text is **grammatically correct** and sounds **realistic** for human speech or dialogue as written (e.g., plausible dialogue, expressive utterances).
    * It must **NOT** sound forced, nonsensical, overly exaggerated beyond plausibility, or contain cues that contradict each other.
    * The enhancement must integrate **smoothly and coherently**, logically fitting the context and character (if implied). The intended paralinguistic effect should be reasonably clear from the **standard textual cues within the speech**.
    * **Prioritize realism and coherence over maximizing the number or intensity of cues.** If an enhancement feels unnatural using standard cues, choose a simpler or different one.
    * **NO OMMISSION OF ANY TEXT** Do not remove non-paralinguistic text from the **text_to_synthesize**.
    * **NO TEXTUAL MARKERS** Do not add * or ** or () characters to the text.
    * **NO SINGLE LETTER HYPHENATIONS** Do not add single letter hyphenations like Y-O-U. Hyphenations should always be natural syllable stressing.
6. **Final Check:** Read the rewritten text aloud. Does it sound like something a person might realistically say? Is the intended paralinguistic cue clear *from the text itself using standard conventions*? Does it pose a relevant challenge for TTS rendering based on these embedded cues?

** In the **tts_synthesis_diversity** field, you MUST provide detailed reasoning covering:
1. The specific paralinguistic enhancement technique you want to use and how you will apply it(which specific cue you will add) within the spoken text using standard conventions. You can choose to add one or two cues.
2. Comment on the novelty of the cue/cues you will add, if this is in one of the examples provided within the technique description above, or you came up with your own. Novel cues are preferred but not required.
3. How the change will **SIGNIFICANTLY** enhance the paralinguistic challenge for TTS, specifying the intended vocal effect after the change.
4. **Crucially, justify *why* the rewritten text remains grammatically correct, sounds realistic for speech/dialogue, and integrates smoothly. Explain how coherence and logical flow were maintained.** Address the critical constraint directly, including adherence to the rule about avoiding meta-textual cues and non-standard formatting.
"""

Figure 8: Example depth-refinement for emotions category. In refinement 1, we add narrative text and contrasting emotion to the one already present. This emotion is further intensified in refinement 2. In the final refinement, we add another contrasting emotion, to have three emotional states, disbelief → fury → heartbreak.

## A.4 Category 4: Emotions

**Breadth Expansion**    The initial dataset of 20 samples from BASE-TTS, while covering foundational emotions like joy, anger, and sadness, exhibited limitations in emotional granularity and contextual depth necessary for comprehensive TTS evaluation. It predominantly featured strong, primary emotions and lacked sufficient diversity in more nuanced states such as sarcasm, envy, resignation, or complex blends like bitter-sweetness. To address these gaps, an additional set of 50 samples was curated using **Claude 3.7 Sonnet**, specifically designed to significantly expand the emotional palette and sentence structural variety. This augmented set incorporates a wider spectrum of subtle and complex affective states, embedded within richer narrative contexts that provide stronger implicit cues for the target prosodic realization. The resulting 70-sample dataset thus offers enhanced evaluative robustness, enabling a more rigorous assessment of a TTS system's ability to synthesize expressive dialogues. The prompt used for breadth expansion is:

```
Consider the below set of 20 samples. This set belongs to the "Emotions" category
    and you will use it to create an extremely diverse set for evaluation of TTS
    systems, where the systems have to synthesize the text corresponding to **
    text_to_synthesize**.
This category assesses whether the system expresses emotions naturally, using
    variations in pitch, loudness, and rhythm. A good TTS system should reflect
    emotions like excitement, sadness, or frustration(and others) as they appear
    within quotes"" in **text_to_synthesize**, and there are many other emotions/
    corresponding context that can be used to evaluate a TTS system which may not
    be covered in the 20 samples.
Your goal is to generate 50 more samples belonging to this category. You will do
    this in the following step-by-step manner:
1. You will analyze the 20 samples carefully.
1.a. Reason deeply about the types of emotions covered by this set.
1.b Reason deeply about the **text_to_synthesize** structures present, like the
    placement of the quoted dialogue, the number of words inside the quoted
    dialogue, the number of words outside the quoted dialogue, the number of quoted
     dialogues, and the grammatical structure of the texts.
2. Now, you will think long about what this set is **MISSING**, specifically, the
    various types of emotions that exist in the **COMPLETE** set of emotions
    present in quoted texts/dialogues, but are not present in this set, **AND**
    will be great to test emotion expressiveness synthesis ability of TTS systems.
3. Finally, you will create additional 50 samples, that expand the current 20 sample
     set in terms of **DIVERSITY**, as you are doing a breadth-wise evolution of
    this 20 set.
```

```
The main goals for the set are:
1. The emotion should be strongly inferrable from the context, as the TTS system
    will not be explicitly provided(like with a special tag) the emotion needed for
     the dialogue.
2. **All text_to_synthesize that will be generated in the 50 samples, should form a
    fluent sentence that could naturally be seen in a story book, or when someone
    is narrating something.**
3. There should be diversity in terms of placement position of quoted dialogue, and
    the length of the context.
4. Each sample must have ATMOST 2 quoted dialogues, in most cases just 1 quoted
    dialogue.(verify this with the initial 20 set)
5. The 50 samples will follow the same JSONL format as the 20 samples, **BUT** no
    sample in the 50 set should be similar with each other, or the original 20 set.
6. You will not create texts with * word * or ** word ** or add text inside
    parenthesis () to indicate something.
7. The context plays a big role in emphasizing what the emotion should be, so
    generate rich context in all cases.
8. All quoted dialogues must have atleast 5 words.
9. Do not use paralinguistic cues or punctuations exc

Now, you are given the 20 sample set, after this, think deeply and create the 50
    foreign words set.

'''jsonl
<now the 20 samples were provided in jsonl format>
'''
```

**Depth Refinement**  We leverage the depth refinements to test TTS systems on more than just
producing single, unchanging emotions. This approach checks two key things: first, how well the
system changes its emotional expression when the text suggests a shift (like moving from happy
to sad), and second, how realistically it can keep an emotion going or make it stronger within a
single piece of dialogue, just like people do when they speak naturally. We refine the base samples to
introduce increased complexity, primarily through two mechanisms: either incorporating a distinct
contrasting emotional state-often signaled via brief preceding or subsequent narrative cues-or by
deepening and intensifying an existing emotion within a specific dialogue segment, thereby extending
the utterance. Emphasis was placed on ensuring the plausibility of these emotional arcs through
natural narrative flow, and matching the existing language style of the text to ensure overly formal
language is not introduced where it does not fit. Refer to refinements in Figure 8 and the prompt used
is:

```
    complex_prompt_method = """
This **text_to_synthesize** belongs to the "Emotions" category.
This category evaluates whether a TTS system clearly and naturally expresses
    fundamental emotions using prosodic, rhythmic and intonational variation. The
    aim is to generate **unambiguously evaluable** dialogues testing mastery of
    distinct emotional expression, including sustained emotion within utterances.

# Your Task: Evolve the **text_to_synthesize** through the following steps:
1. **Identify ALL quoted dialogues** and capture their length in the **
    text_to_synthesize**.
2. **Choose one of the following methods to evolve the text:**
    * **Method A - DEEPENING:**
        * **Condition:** The length of words in ATLEAST ONE of the quoted dialogues
            is **LESS THAN OR EQUAL TO 20 words**. If such a quote exists, identify
            it as a **SHORT QUOTE**. (If multiple SHORT QUOTES exist, choose the one
             where deepening the existing emotion feels most natural.)
        * Append new dialogue *directly within the same quotes* as the chosen **
            SHORT QUOTE**, extending the character's speech to reflect the
            INTENSIFIED/DEEPER continuation of the emotion that is already present
            in that **SHORT QUOTE**.
        * Do **NOT** add new **narrative text** before this appended dialogue; only
            append **INSIDE** the chosen **SHORT QUOTE**.
```

* The new dialogue will be a continuation of existing dialgue, and can
            either continue the sentence in the dialogue by adding before full stop
            (.), or start a new sentence(but within the same quotes). **BE CREATIVE
            HERE**
    * **Method B - CONTRASTING:**
        * **Condition:** If Method A is not applicable (no quotes are <= 20 words),
            use Method B.
        * Add descriptive, natural **narrative text** (max 10 words) **either before
            (prefix) or after (suffix)** the original **text_to_synthesize**. This
            **narrative text** should also **BRIDGE** plausibly with the **quoted
            dialogue**.
        * Follow/Precede this **narrative text** with **ONE new, separate quoted
            dialogue** ("...") expressing the **CONTRASTING** core emotion.
        * The text within this **new contrasting quote** should be **at least 5
            words** long and **at most 15 words** long.
        * For clarity, 4 orders are allowed in this method:
            * **Order 1:** **quoted dialogue** + **unquoted narrative text** + <**
                text_to_synthesize**>
            * **Order 2:** **unquoted narrative text** + **quoted dialogue** + <**
                text_to_synthesize**>
            * **Order 3:** <**text_to_synthesize**> + **quoted dialogue** + **
                unquoted narrative text**
            * **Order 4:** <**text_to_synthesize**> + **unquoted narrative text** +
                **quoted dialogue**

3. The overall evolution must:
    * Introduce **ONE clearly identifiable CORE EMOTION** (focus on Joy, Sadness,
        Anger, Fear, Surprise, Disgust)(Method B) OR a significant intensification (
        deepening) of the existing emotion(Method A).
    * Flow **naturally and plausibly** (PRIORITY #1).
    * You **WILL** choose one of the characters from the narration(if there are
        multiple characters) and add the dialogue for that character in case of
        Method B. You **WILL NOT** create a new character in the narration.
    * You have to realize the tone of the narration, and continue the dialogue in
        the same tone. Do not use overly formal language where it does not fit, or
        too casual where it does not fit. Its **IMPERATIVE** to analyze the tone
        correctly.

# IMPORTANT CONSTRAINTS & GUIDELINES (Apply Always):

1. **MAXIMUM EVALUABLE CLARITY (HIGH PRIORITY):** Emotion must be **clearly
    identifiable** and **distinct** based on the pretext/context(if added like in
    Method B).
2. **EMOTIONAL CUE:**(Only for Method B) Ensure strong textual cues are used to lead
    upto the emotion(quoted dialogue), but cues should not be too artificial or
    poetic, they should be natural.
3. **NATURAL & SPOKEN STYLE:** Use authentic, spoken language. Avoid literary prose
    and use language that fits the tone of the narration.
4. **PLAUSIBILITY & COHERENCE:** Evolution must be believable. Addition must feel
    grounded.
5. **STRUCTURAL RULES:** Follow Method A/B strictly. Do not change original text's
    **EMOTION**.
6. **NO TEXT MARKERS:** **DO NOT** add additional text markers like the characters
    "*" or "**" or any other text markers in the **rewritten_text_to_synthesize**.
7. **Transitional Text:** The transition between narrative text and quoted dialogue
    **MUST** be natural.
8. **Grammatical Correctness:** The **rewritten_text_to_synthesize** must be
    grammatically correct.
# Output Explanation Fields:

## In the **tts_synthesis_diversity** field, you must provide the following
    information:
    1. **Method Choice Rationale:** Explain *why* you chose Method A or Method B and
        subsequently if the added text will be emotionally **DEEPENING** or **

```
        CONTRASTING**. (If Method A, state which SHORT QUOTE was chosen if multiple
        existed).
   2. *Recognize the tone of the narration, and mention it here, this is the tone
        (**with slight variations accepted**) that you have to continue in the
        rewritten dialogue.*
   3. *(Only if Method B was used):* Think about plausible **narrative text** and
        **quoted dialogue** to create emotional arcs for all 4 orders.
        - 3.1 **Order 1** possible emotional arc.
        - 3.2 **Order 2** possible emotional arc.
        - 3.3 **Order 3** possible emotional arc.
        - 3.4 **Order 4** possible emotional arc.
        - 3.5 Identify if applying **Order 2** or **Order 3** may result in **
             rewritten_text_to_synthesize** containing two **consecutive quoted
             dialogues**. If **text_to_synthesize** contains quoted dialogues at
             boundaries this may happen and you will **ELIMINATE** problematic orders.

        - 3.6 Based on the analysis of all 4 emotional arcs and eliminating the
             invalid orders, choose the best out of the valid orders and justify your
             choice.
   4. The **specific CORE EMOTION** we can add(Method B) or significantly deepen(
        Method A, this is where you will specify which emotion is being deepened).
   5. *(Only if Method A was used):* Specify the **continuation** or **new sentence
        ** that you will add inside the quotes of **SHORT QUOTE** to intensify the
        emotion.
   6. *(Only if Method B was used):* The **narrative text** and the **quoted
        dialogue** you will add according to the best order you chose in point 3.6.
   7. *(Only if Method B was used):* Identify any **MINOR PARAPHRASING** of the **
        text_to_synthesize** to accomodate **narrative text** and **quoted dialogue
        ** you added in point 6.
   8. Ensure that the **rewritten_text_to_synthesize** to be created follows **
        IMPORTANT CONSTRAINTS & GUIDELINES**, it does not contain any "*" or "**"
        markers, and is **grammatically correct.**
   9. Ensure that all the decisions you made make for a better TTS evaluation than
        the original **text_to_synthesize** for emotional expressiveness.
  10. Be CAREFUL with the handling of quotes in the output json object, always
        escape the quotes in the **rewritten_text_to_synthesize** field in the
        output json object. Every opened quote must have a closing quote.
```

## A.5   Category 5: Syntactic Complexity

```
┌─────────────────────────────────────────────────────────────────────────────┐
│ Why the council rejected the proposal, despite acknowledging its potential benefits and the │
│                    strong public support it garnered, remains unclear.                      │
└─────────────────────────────────────────────────────────────────────────────┘
                                        ↓
┌─────────────────────────────────────────────────────────────────────────────┐
│ Why the council rejected the proposal, a project detailed extensively, despite acknowledging │
│  its potential benefits and the strong public support it was projected to garner, remains    │
│                                        unclear.                                              │
└─────────────────────────────────────────────────────────────────────────────┘
                                        ↓
┌─────────────────────────────────────────────────────────────────────────────┐
│ Why the council rejected the proposal, a project detailed extensively, and how swiftly they  │
│  dismissed alternatives, despite acknowledging its potential benefits and the strong public  │
│                    support it was projected to garner, remains unclear.                      │
└─────────────────────────────────────────────────────────────────────────────┘
                                        ↓
┌─────────────────────────────────────────────────────────────────────────────┐
│ Why the council, expected to object to flawed plans, rejected the proposal, an object detailed │
│  extensively, and how swiftly they dismissed alternatives, despite acknowledging its potential │
│        benefits and the strong public support it was projected to garner, remains unclear.     │
└─────────────────────────────────────────────────────────────────────────────┘
```

Figure 9: Example depth-refinement for syntactic complexity category. Initially, it presents a multi-clause sentence requiring clear phrasing. Subsequent refinements introduce further complexity: first with an embedded appositive, then a coordinated dependent clause increasing structural intricacy. The final stage incorporates an additional non-restrictive clause and, critically, the homograph "object"-used as both a verb and a noun-to assess the TTS's ability to disambiguate via pronunciation and stress within a highly embedded structure.

**Breadth Expansion**  The initial 20 samples effectively tested TTS prosody on deep center-embedding and long subject-verb dependencies but notably omitted other crucial structures reliant on prosody, such as inversion, cleft sentences, ellipsis (gapping), complex clausal subjects (Wh-/That-/gerunds), and nuanced punctuation cues (semicolons, dashes). The 50 samples curated through breadth expansion using **Gemini-2.5-pro** rectify these specific omissions by introducing robust examples across these categories, significantly broadening the structural diversity and creating a more comprehensive benchmark for evaluating a TTS system's handling of complex syntax. The resulting 70-sample dataset therefore provides a more robust and syntactically varied test suite for assessing the prosodic competence of TTS systems when faced with intricate grammatical structures. The breadth expansion prompt is as follows:

```
    Consider the below set of 20 samples. This set belongs to the "Syntactic
        Complexity" category and you will use it to create an extremely diverse set
        for evaluation of TTS systems, where the systems have to synthesize the text
        corresponding to **text_to_synthesize**.
This category evaluates how well the system uses **prosody (pausing, phrasing,
    intonation, stress)** to make complex sentence structures easily understandable.
     It tests if the TTS clearly conveys the intended grammatical relationships,
    especially with nested clauses, ambiguities, or long dependencies. There are
    other forms of syntactic complexity that can be used to evaluate a TTS system
    which may not be covered in the 20 samples.
Your goal is to generate 50 more samples belonging to this category. You will do
    this in the following step-by-step manner:
1. You will analyze the 20 samples carefully.
1.a. Reason deeply about the types of syntactic complexities present.
2. Now, you will think long about what this set is **MISSING**, specifically, the
    various types of syntactic complexities that exist in the **COMPLETE** set
    english sentences, but are not present in this set, **AND** will be great to
    test complex grammar synthesis ability of TTS systems.
3. Finally, you will create additional 50 samples, that expand the current 20 sample
     set in terms of **DIVERSITY**, as you are doing a breadth-wise evolution of
    this 20 set.

The main goals for the set are:
1. We want to include all types of syntactic complexities, that are **ATLEAST** as
    complex as the ones present in the 20-sample set.
2. The 50 samples will follow the same JSONL format as the 20 samples, **BUT** no
    sample in the 50 set should have phrasing that gives raise to the **SAME**
    syntactic complexity as the 20 sample set. Our goal is to create a **DIVERSE**
    set.
3. You will not create texts with * word * or ** word ** or add text inside
    parenthesis () to indicate something.

Now, you are given the 20 sample set, after this, think deeply and create the 50
    syntactic complexity set.
```jsonl
<now the 20 samples were provided in jsonl format>
```
```

**Depth Refinement**  While breadth-wise expansion verifies coverage across diverse syntactic phenomena, depth-wise refinement is crucial for assessing a TTS system's robustness and performance scalability when faced with escalating grammatical intricacy. This approach tests the system's ability to manage compounded syntactic load and maintain prosodic coherence under increasing structural demands, rather than merely handling isolated complexities. Our refinement strategy involved iteratively enhancing base complex sentences by applying targeted syntactic transformations-such as introducing complex coordination, structural reordering (e.g., fronting, passivization impacting dependencies), complicating ellipsis, or adding layered subordination-while strictly enforcing constraints on grammatical correctness and naturalness. The refinement is also encouraged to add two words that are homographs of each other if it fits naturally in the context, this tests the ability of the TTS to disambiguate the meaning from context and correctly pronounce it. The resulting dataset provides a graded challenge, enabling evaluation of how TTS prosody adapts to and conveys meaning

across incrementally complex, yet plausible, sentence structures. Refer to Figure 9 for an example refinement, and the prompt used is:

```
complex_prompt_method = """
You are tasked with enhancing the syntactic complexity of the provided **
    text_to_synthesize**, which belongs to the "Syntactic Complexity" category.
    This category tests a TTS system's ability to render complex grammar clearly
    through prosody, pauses, intonation and stress patterns.

Your specific task is to rewrite the **text_to_synthesize** by performing the
    following steps:

1. **Analyze the existing sentence structure** and identify opportunities for
    enhancement.
2. **Enhance syntactic complexity using ONE appropriate technique.** Select a
    technique that fits naturally and logically, aiming for variety across
    evolutions. **Give special consideration to techniques beyond simple clause/
    phrase addition, such as:**
   * Introducing or complicating **coordination** (of long/dissimilar phrases or
        clauses).
   * Introducing or complicating **gapping/ellipsis** (omission of repeated
        elements, esp. verbs).
   * Restructuring to create **scope ambiguity** resolvable by prosody (e.g.,
        involving negation, quantifiers).
   * Employing **structural reordering** (e.g., complex fronting, heavy subject/
        object shifts, passivization impacting dependencies).
   * Adding/elaborating **subordinate elements** (nested clauses, complex
        appositives, participial phrases) - *use if other options don't fit
        naturally*.
   * Increasing **modifier density** or creating **longer-distance dependencies**
        through restructuring.
   * Leveraging **punctuation** (colons, semicolons) to structure complex
        relationships.
3. **Optional Homograph Integration:**
   * **IF AND ONLY IF** it fits **perfectly naturally** within the enhancement you
        are making, you MAY include **two words that are homographs** of each other
        (different standard pronunciations, contextually unambiguous).
   * **Do NOT use special formatting** around homographs in the **
        rewritten_text_to_synthesize**.
4. **Complexity Goal:** The technique should aim to *appropriately enhance* the
    overall syntactic complexity in a **meaningful structural way**, without
    sacrificing clarity, naturalness, or grammatical correctness.
5. **CRITICAL CONSTRAINT - Length, Naturalness, Grammar & Coherence:**
   * You will increase the length of the **text_to_synthesize** by **ATLEAST 2
        words** and **ATMOST 6 words**.
   * Your **absolute priority** is ensuring the rewritten text is **grammatically
        flawless**, sounds **natural** for complex written English, and flows
        logically.
   * It must **NOT** sound forced, artificially constructed, overly convoluted,
        ambiguous due to structure (unless ambiguity is the intended challenge,
        solvable by prosody), or like a mere linguistic puzzle.
   * The enhancement must integrate **smoothly and coherently**, contributing
        logically or structurally to the sentence. Avoid awkward interruptions or
        obscuring core grammatical relationships excessively.
   * **Prioritize naturalness/grammar over maximizing complexity.** If an
        enhancement feels forced, choose a simpler or different one.
6. **Final Check:** Read the rewritten sentence aloud. Does it flow logically? Is
    the intended structure parseable? Does it introduce a relevant syntactic
    challenge for TTS prosody?

** In the **tts_synthesis_diversity** field, you MUST provide detailed reasoning
    covering:
1. The specific complexity enhancement technique you used (referencing the examples
    if applicable) and how you applied it.
```

```
2. **If homographs were used:** Identify them and explain how the context makes
   their pronunciation unambiguous. **If not used, state that.**
3. How the change **ENHANCES** syntactic complexity, focusing on the **specific
   structural challenge** it poses for TTS prosody, pausing, intonation and stress
    patterns(e.g., handling ellipsis gap, coordinating long phrases, resolving
   scope, pausing timing).
4. **Crucially, justify *why* the rewritten sentence remains grammatically correct,
   sounds natural, and integrates smoothly despite the enhancement. Explain how
   coherence and logical flow were maintained.** Address the critical constraint
   directly.
5. Ensure the text does not contain any extra characters like ', *, ** , (), etc.
"""
```

## A.6  Category 6: Complex Pronunciation

Figure 10: Example depth-refinement for complex pronunciation category. Initially, we have standard version strings and symbols. The first refinement introduces a significantly more complex, multi-component version string and a new foreign currency (¥). Then, it adds distinct challenge of correctly verbalizing hexadecimal numbers (e.g., "0x") and technical abbreviations like "rev." (revision). Finally, the third refinement assesses the system's ability to interpret and articulate numerical uncertainty or ranges indicated by the "±" symbol.

**Breadth Expansion**   We create this category from scratch by prompting **Claude 3.7 Sonnet** to generate 60 samples, 10 from each of the following 6 categories, (i) Numerals and Currencies, (ii) Dates and Times, (iii) Emails, URLs and Passwords, (iv) Addresses and Location references, (v) STEM Notations and equations (vi) Mixed Acronyms/Initialisms. In addition to this, we add 5 short tongue-twisters that are repeated many times. For example, the "The Sixth Sick Sheikh's Sixth Sheep's Sick" tongue twister repeated 6 times. The prompt used is:

```
    Your goal is to generate a dataset of 60 samples. Each sample is a json line, so
        the dataset should be returned as a jsonl. Each json contains 2 keys:
        category: This will be set to "Pronunciation" text_to_synthesize: This is
        the key field that you will populate with great diversity.
The goal of this dataset is to evaluate TTS systems where they have to synthesize
    whatever text you populate in text_to_synthesize. This is the "Pronounciations"
     category, we want to create samples with the following outline. 10 samples for
     each category(6 categories): 1. Text with currency and numerals in different
    formats 2. Text with dates and time-stamps in different formats 3. Texts with
    email addressess, passwords and urls in different formats. 4. Texts with
    complex street addresses or location references. 5. Texts with terms from the
    STEM field. 6. Texts that have both an initialism and acronym These are the two
    main goals of this dataset:
1. Breadth: It should cover huge diversity in sentence structure, sentence length.
2. Depth: This dataset stress-tests the pronounciation capability of TTS systems.
    Before generating the dataset, reason step-by-step and deeply on how you will
    cover the breadth and depth aspects, then give 60 jsonl lines
```

**Depth Refinement**    In this category, the goal of depth refinement is to progressively add complex, hard-to-pronounce elements to an utterance, while keeping them within the same sub-category as the original. With each refinement, we aim to increase the density of such elements. To avoid repeating the same pronunciation challenges, we prompt the refining LLM to suggest three novel ways to introduce complexity - distinct from what's already present. Specifically, we ask for elements that are likely to challenge TTS systems, even if other parts are rendered correctly. This approach consistently produces utterances with multiple challenging components that TTS systems may struggle to synthesize. We do not apply this strategy for the tounge-twisters and keep them as is. An example refinement is given in Figure 10 and the prompt used is:

```
complex_prompt_method = """
   The **text_to_synthesize** belongs to the "Pronunciation" category.
   This category evaluates how well the system pronounces non-trival words,
       numerals and other special characters.
   Your primary goal is to **increase the pronunciation complexity** of **
       text_to_synthesize**, creating a **rewritten_text_to_synthesize**.

   First, the **text_to_synthesize** will fall into one of these 6 methods/
       categories:
   1. **Numerals & Currency:** Focuses on the presence of numbers and currencies in
        varied formats.
   2. **Dates & Times:** Contains dates, times, durations, and time zones in varied
        formats.
   3. **Emails, URLs & Passwords:** Features communication/resource locators (
       emails, URLs, passwords, IPs, phone numbers, etc.).
   4. **Addresses & Locations:** Describes physical locations (street addresses
       with components like numbers/suffixes/directions/abbreviations, coordinates,
       etc.).
   5. **STEM Notations:** Characterized by scientific/math formulas, complex
       notations, specialized units, etc.
   6. **Mixed Acronyms/Initialisms:** Includes **BOTH** spelled-out abbreviations (
       letter-by-letter) and pronounced ones (as words) in the same sentence.

   The provided **text_to_synthesize** belongs to **Category {{{
       pronunciation_sub_category}}}**. Your evolution must remain focused within
       this category **WHILE** ensuring that the rewritten_text_to_synthesize will
       not fall into any other category.

   Now, **evolve the complexity**:
   - Modify the **text_to_synthesize** by introducing **NOVEL** elements that will
       increase the pronunciation complexity of the text.
   - You may paraphrase the original text *slightly* for natural flow, but **
       CRITICAL: ensure the specific complex elements** from the **
       text_to_synthesize** are preserved in the **rewritten_text_to_synthesize**
       unless they are being directly merged or replaced by the new, more complex
       elements.
   - Focus on introducing **different facets, new dimensions, or more intricate
       variations** of complex elements to do the evolution.
   - **CRITICAL:** **Do NOT simply add a different, unrelated element type, even if
        it falls under the same broad category number.** The goal is to make the *
       existing type* of challenge harder, not to dilute it with a separate
       challenge.

   In the **tts_synthesis_diversity** field, provide your analysis:
   1. Briefly explain the **complex elements** present in the **text_to_synthesize
       ** that will particularly challenge the TTS system, confirming its alignment
        with Category {{{pronunciation_sub_category}}}.
   2. Reason about **MULTIPLE DIFFERENT** **complex elements** we may introduce to
       the **text_to_synthesize** that will increase the pronunciation complexity.
       2.1 For each candidate **complex element**, reason how the TTS system may
           fail to pronounce it **EVEN IF** all existing **complex elements**
           present in the **text_to_synthesize** are pronounced correctly.
   3. Reason which **complex elements** you came up with in step 2 will be the most
       effective to increase the pronunciation complexity of the text in a **novel
```

```
            ** way. Select **EXACTLY 1 element** and call it the **chosen complex
            element**.
      4. What will be the ideal pronunciation for the **chosen complex element** by
         the TTS system?
      5. Identify the best way to introduce the **chosen complex element** in the **
         text_to_synthesize** to increase the pronunciation complexity of the text.
      6. Confirm that the introduction of the **chosen complex element** won't make
         the text artificial or unnatural or domain-inappropriate.
      """
```

## A.7 Final dataset statistics

Refer to Table 4 for final category-wise statistics.

Table 4: Final Dataset Statistics

| Category | No. Characters | | | No. Words | | | Audio Length (s) | | |
|---|---|---|---|---|---|---|---|---|---|
| | Min | Avg | Max | Min | Avg | Max | Min | Avg | Max |
| Questions | 16 | 248.22 | 701 | 3 | 41.61 | 120 | 0.90 | 15.04 | 48.45 |
| Foreign Words | 71 | 136.85 | 242 | 9 | 21.77 | 39 | 4.80 | 9.07 | 16.20 |
| Paralinguistics | 28 | 127.36 | 319 | 5 | 19.30 | 49 | 2.15 | 9.23 | 22.30 |
| Emotions | 102 | 340.04 | 676 | 18 | 57.58 | 107 | 6.15 | 21.80 | 45.20 |
| Syntactic Complexity | 45 | 194.71 | 366 | 8 | 28.23 | 64 | 3.25 | 12.53 | 23.60 |
| Complex Pronunciation | 104 | 260.35 | 920 | 8 | 35.28 | 139 | 8.45 | 25.56 | 94.70 |
| Overall | 16 | 217.02 | 920 | 3 | 33.93 | 139 | 0.90 | 15.32 | 94.70 |

# B   LALM benchmarking for audio understanding

This section shows results from 4 audio understanding benchmarks. These results demonstrate Gemini-2.5-pro's strong capabilities and provide justification for using it as a judge model to assess TTS output.

MMAU [32] is an audio-reasoning benchmark, testing audio understanding models for reasoning across 3 categories, Speech, Music and Sounds. We run the evaluation ourselves on the test-mini subset of 1,000 samples for top closed-source LALM models, and the results are summarized in Table 5.

Table 5: LALM performance on Audio Reasoning MMAU benchmark, test-mini subet of 1000 samples

| Model | Test-mini score ↑ |
|---|---|
| Gemini 2.5 Pro | **68.60** |
| Gemini 2.5 Flash | 65.20 |
| Gemini 2.0 Flash | 62.10 |
| Gpt-4o-audio | 59.20 |
| Gpt-4o-mini-audio | 59.80 |

StressTest [47] is a benchmark designed to assess how well LALMs understand emphasis in spoken language. It consists of recordings by a professional voice actor who repeats the same sentence while stressing different words, producing multiple versions that vary only by emphasis and subsequently in their intended meaning. This setup tests whether a model can interpret how meaning changes depending on which word is stressed.

We adapt this benchmark into a pairwise comparison task, where the LALMs must identify which of two audio samples matches a text in which a specific word is emphasized, this closely matches our judger setting and results in 266 samples. Performance is measured by accuracy, with random

guessing yielding a 50% baseline. The resulting scores are shown in table 6. Gemini-2.5-pro achieves an accuracy of 100%, which is possible since the audios are very clean as they are recorded by a professional actor, but other models still struggle. This shows that Gemini-2.5-pro can effectively recognize stress/emphasis in audio, which makes it useful as a judge for our benchmark.

Table 6: LALM performance on modified StressTest benchmark

| Model | Accuracy ↑ |
|-------|-----------|
| Gemini 2.5 Pro | **100%** |
| Gemini 2.5 Flash | 93.23% |
| Gemini 2.0 Flash | 81.20% |
| Gpt-4o-audio | 66.66% |
| Gpt-4o-mini-audio | 56.76% |

Next, we refer to the StepEval-Audio-Paralinguistic benchmark proposed with the Step-Audio 2 model [43], this benchmark evaluates LALM capability to understand paralinguistic information. We compare Gemini-2.5-pro against Step-Audio 2 on 6 categories relevant to our proposed TTS evaluation: Emotion, Pitch, Rhythm, Speed, Style and Vocal Sounds. The results are in table 7.

Table 7: Gemini-2.5-pro performance on StepEval-Audio-Paralinguistic

| Model | Average ↑ | Emotion | Pitch | Rhythm | Speed | Style | Vocal |
|-------|-----------|---------|-------|--------|-------|-------|-------|
| Step-Audio-2(reported) | 77.33 | 72 | 78 | 70 | 78 | 84 | 82 |
| Gemini 2.5 Pro | **82.00** | 86 | 84 | 74 | 84 | 80 | 84 |

Finally, we refer to the EmoNet-Voice [33] benchmark. This work assessed models on nuanced emotion recognition across 40 distinct emotional intensity labels. Their results show that Gemini 2.5 Pro is the top-performing general-purpose model in terms of correlation with expert human annotations (corr=0.417). This is on par with their own model trained specifically on an emotion recognition dataset (corr=0.421). Notably, this level of correlation is significant given the low inter-rater agreement among human annotators themselves (Cronbach's $\alpha$=0.14), this low inter-rater agreement is due to the subjectivity of audio and is noted in Appendix D in our paper.

## C  Evaluation-related Details

### C.1  Hyper-parameters

**Data Depth Refinement:**    Gemini 2.5 Pro is prompted with $temperature = 1.0$ for creativity, $top\_p = 0.9$ and $max\_output\_tokens = 16384$ when doing depth refinement for 3 steps.

**Audio Generation:**    Closed source TTS models like Aura-2, Eleven Multilingual v2, HumeAI and gpt-4o-mini-tts do not support a temperature parameter. For Sesame1B, Qwen2.5 Omni, gpt-4o-mini-audio-preview and gpt-4o-audio-preview we use a $temperature = 1.0$, and for orpheus tts, we use the recommended values of $temperature = 0.6$ and $top\_p$=0.8. We set the maximum output tokens to 8192 and ensure that for no system, the audio is being clipped. However, for MiniCPM, the audio is automatically clipped at 44 seconds, and there does not appear to be an exposed parameter to extend this limit. Suno-ai's Bark only performs well up to 13, the official repository provides a recipe for generating longer audio by splitting the input into multiple sentences and concatenating the outputs - we adopt this method. Tortoise-TTS audio is clipped at around 27 seconds, even after setting `max_mel_tokens` to its maximum value of 600. For Zyphra's Zonos model, we follow the inference code from the GitHub repository and use `assets/exampleaudio.mp3` as the prompt audio for voice cloning.

**Judger LALMs:**    For the judger, we set $temperature = 0.0$ for reproducibility, we find that while Gemini results are not deterministic even after setting a value of $0.0$, the final win-rate does not change significantly across runs, $< 1\%$ change to be specific. The max output length is set to 131072 for the thinking models (Gemini 2.5 series), and 16384 for other judgers we use for ablation.

## C.2 Prompts for Audio Generation

From the description map presented below, we select and send the description relevant to the specific category when using **Strong Prompting** with Hume AI and gpt-4o-mini-tts:

```
ALL_DESCRIPTIONS = {
    "Emotions": "Emotional expressiveness: Ensure a clear and distinct transition
        between quoted dialogues and narrative text. Deliver the quoted dialogues
        with high emotional expressiveness.",
    "Paralinguistics": "Paralinguistical cues: Express interjections, onomatopoeia,
        capitalization, vowel elongation, hyphenation/syllable stress, stuttering
        and pacing cues(elipses, punctuations, etc.) naturally and realistically.",
    "Syntactic Complexity": "Syntactical Complexity: Maintain clarity in complex
        sentence structures through appropriate prosody, pausing and stress to
        convey the intended meaning of the sentence very clearly. Handle homographs
        with appropriate pronunciation.",
    "Foreign Words": "Foreign words: Pronounce foreign words and phrases with their
        appropriate pronunciation or anglicized version, sound like a natural bi-
        lingual speaker doing smooth code-switching.",
    "Questions": "Questions: Apply the appropriate intonation pattern for
        interrogative sentences(questions) and declarative sentences.",
    "Pronunciation": "Complex Pronunciation: Pronounce currency, numerals, emails,
        passwords, urls, dates, times, phone numbers, street addresses, equations,
        initialisms, acronyms, tounge twisters(speak fast while maintaining clarity)
        , etc. with precision, clarity and case-sensitivity wherever applicable."
}
```

Following are the templates we use for normal prompting and strong prompting scenarios with LALMs like Qwen 2.5 Omni, gpt-4o-mini-audio-preview and gpt-4o-audio-preview.

**Normal Prompt**:

```
USER_MESSAGE_DEFAULT_TEMPLATE = """
    Your goal is to synthesize speech that exactly matches the text under **
        text_to_synthesize** tag.
    You will be provided with the **text_to_synthesize**, after that generate **ONLY
        ** the **VERBATIM** speech matching the text. Do not add any additional
        information or text in your response.
    ***text_to_synthesize***:
    {{{text_to_synthesize}}}
    """
```

**Strong Prompt**:

```
USER_MESSAGE_STRONG_TEMPLATE = """
    Your goal is to synthesize speech that exactly matches the text under **
        text_to_synthesize** tag.
    The generation has to be human-like and realistic. To excel in this task, you
        must pay attention to the following aspect of the text:
    {{{descriptions}}}
    Now, you will be provided with the **text_to_synthesize**, after that generate
        **ONLY** the **VERBATIM** speech matching the text. Do not add any
        additional information or text in your response.
    ***text_to_synthesize***:
    {{{text_to_synthesize}}}
    """
```

The {{{descriptions}}} placeholder is replaced with the specific description of that category, as mentioned in the ALL_DESCRIPTIONS map.

## C.3 Prompts for Judger and category-wise evaluation criteria

The judger is provided with the following prompt template:

```
USER_MESSAGE_WIN_RATE = """Your goal is to judge two TTS(text-to-speech) systems and
    analyze which system synthesizes speech corresponding to a particular text
    better than the other one and determine the winner based on the scoring
    criterion.
You will rate each system a score between 0 and 3 based on how well it
    synthesizes speech corresponding to a particular text called **
    text_to_synthesize**, then do their comparative analysis and provide your
    final judgement.
A good system will generate speech that sounds realistic and human-like, and it
    captures the specific nuances of the text.

You will be provided with the **text_to_synthesize** which is the text both TTS
    systems had to synthesize,
the **text_category** and the **evaluation_criterion** corresponding to the **
    text_category**, in which you will be made aware of the **evaluation
    dimension** you will focus on, and the **scoring criteria** you will use to
    score the TTS systems.
You will also be provided with the **output_format**, which dictates the format
    of the output you need to follow as a judger.
Finally, you will be provided with the synthesized speech from the TTS system 1
    **synthesized_speech_1** and then from TTS system 2 **synthesized_speech_2**.

**text_to_synthesize**
{{{text_to_synthesize}}}

**text_category**
{{{text_category}}}

**evaluation_criterion**
{{{evaluation_criterion}}}

NOTE: If the generated speech is very poor and does not synthesise the text
    correctly, you will provide a score of 0 to that TTS system.
GLOBAL CONSIDERATIONS(**VERY IMPORTANT FOR COMPARISON**):
    - It is imperative to compare the two systems **ONLY** on the basis of the **
        evaluation_dimension**, that means, you **WILL NOT** let the following
        types of **BIASES** affect your judgement.
        - The acoustical quality of the audio, background noise or clarity.
        - The gender and timbre features of the speaker.
        - Any other factors that are not related to the **evaluation_dimension**.
        - Systems demonstrating exaggerated expressiveness should not be rewarded
            more **UNLESS** those features are relevant to the **
            evaluation_dimension**.
    - Tie-break procedure
        1. If the overall score_1 and score_2 are equal, use this protocol.
        2. For the chosen **evaluation_dimension**, inspect every comparable
            component:
            - Note similarities.
            - Note differences and label each as:
                - Subtle: hardly noticeable to a typical human listener.
                - Significant: clearly influences human perception.
        3. Count the significant differences that benefit each system.
        4. Decision:
            - No significant differences, or counts are equal -> declare a tie.
            - Otherwise -> declare the system with the higher count of
                significant advantages the winner.

**output_format**
You will output a json dictionary as follows:
```json
{
```

```
    "reasoning_system_1": str = Reasoning chain based on the **Reasoning guidelines
        :** for the synthesized speech from TTS system 1.
    "reasoning_system_2": str = Reasoning chain based on the **Reasoning guidelines
        :** for the synthesized speech from TTS system 2, **INDEPENDENT** of the
        performance of TTS system 1.
    "system_comparison": str = Keeping in mind the GLOBAL CONSIDERATIONS, compare
        and contrast the two systems based on your output in reasoning_system_1 and
        reasoning_system_2 and also by analyzing both audios again. Provide very
        fine-grained reasoning for which system won, or if the comparison results in
         an even tie.
    "score_1": int = Your score for the synthesized speech from TTS system 1 between
         0 and 3, based on the **evaluation_criterion** and what you have mentioned
        in reasoning_system_1.
    "score_2": int = Your score for the synthesized speech from TTS system 2 between
         0 and 3, based on the **evaluation_criterion** and what you have mentioned
        in reasoning_system_2.
    "winner": int = The winner of the comparison between TTS system 1 and TTS system
         2. Output 1 if TTS system 1 wins, 2 if TTS system 2 wins, output 0 if this
        will be considered as an even tie.
    }
    - Note: Ensure the json structure is followed and the json output **MUST** be
        parsable without errors.(For example, escape the quotes whereever you add
        them inside a field of the json, all brackets and braces should be correctly
         paired.)

    Now you will be provided with the synthesized speech from the TTS system 1,
        please analyze it carefully.

    **synthesized_speech_1**
    """
```

After the above prompt, we append the audio from System 1, then we have the post audio 1 prompt, this design choice is adopted to provide effective seperation between the two audios.

```
POST_AUDIO_1_MESSAGE = """
Now you will be provided with the synthesized speech from the TTS system 2, please
    analyze it carefully. After that provide the judgment following the **
    output_format** ensuring parsability.
**synthesized_speech_2**
"""
```

The placeholder {{{evaluation_criterion}}} is replaced with the specific criteria for that category, as described in the map below:

```
CATEGORY_TO_CRITERION_MAP = {
        "Questions" : """
        **Evaluation Dimension:**
            - In this category, we want to evaluate the ability of the TTS system to
                 apply correct intonation patterns: Interrogative for questions,
                declarative for statements, etc.
            - Questions usually have a distinct pitch movement, often rising at the
                end in yes/no questions, while wh-questions may have a more neutral
                or falling tone.
            - Statements between questions should have an intonation pattern that
                differentiates them from the questions and makes it clear that it is
                a statement.
            - You have to be careful that texts can have multiple correct intonation
                patterns, so place appropriate weight on parts where intonation is
                not very clear.

        **Example:** "Did you see the message? Well I hope you did. But please tell
            me that you actually did?"
        **Explanation:**
            - There maybe multiple correct patterns to render this speech with, but
                we want to judge if the TTS system has made an attemp at correctly
```

conveying the interrogative intonation for the 2 questinos, and the
declarative intonation for the statement between the questions.

**Note:**
    - The **text_to_synthesize** may contain multiple questions without or
        without the question mark, you have to correctly differentiate
        between the questions and the statements.

**Rating Scale:**
1: All intonation patterns incorrect
2: Some intonation patterns are largely correct but some are incorrect
3: All intonation patterns are correct and convey the question nature
    perfectly

**Reasoning guidelines:**
    1. Mention which parts need to be rendered with interrogative intonation
        and which with declarative intonation.
    2. Carefully list the crucial parts of the speech, the pertinent
        syllables and their precise timestamps.
    3. Analyze the audio multiple times to capture the intonation patters at
        the crucial parts.
    4. Finally, reason deeply and justify how the TTS system has performed
        and applied the intonation patterns at the crucial parts, and then
        what the final score for the TTS system should be.
""" ,

"Emotions" : """
**Evaluation Dimension:**
    - In this category, we want to evaluate the ability of the TTS system to
        express emotions naturally, using variations in pitch, loudness,
        rhythm, etc. and demonstrate tone variations between the quoted
        dialogues and the narrative text.
    - The TTS system has to generate speech as if it is narrating the **
        text_to_synthesize**, which means showing natural and strong
        emotional expressiveness for the quoted dialogues.

**Example:** "Full of joy, he exclaimed: "I can't believe it! This is
    amazing!". But then, a sudden realization dawned on him and he said "
    Okay okay wait wait, I think this may not be such a good idea after all
    ."
**Explanation:**
    - The text inside the first quotes "I can't believe it! This is amazing!"
        should sound excited and joyful, not robotic.
    - The text inside the second quotes "Okay okay wait wait, I think this
        may not be such a good idea after all." should sound disappointed and
        frustrated and this contrasting emotion should be clearly noticeable
        .
    - The narrative between/around the quotes should be distinct than the
        dialogues and should be spoken with the appropriate narrative tone.

**Rating Scale:**
1: Fails to express emotions in the quoted dialogues, and the transition
    between the quoted dialogues and the narrative is flat and not distinct.
2: Synthesises some quoted dialogues with emotions but fails to synthesise
    others, OR, the rendered emotions are not very natural and emphatic, OR,
    the tone bridging quoted dialogues and the narrative text cannot be
    distinguised/is barely discernible.
3: Synthesises all quoted dialogues with natural and emphatic emotions, and
    the tone bridging quoted dialogues and the narrative text is clearly
    distinguishable.

**Note:**
    - The **text_to_synthesize** will not explicitly state the emotion for
        the quoted dialogues, you have to infer that from the context.

**Reasoning guidelines:**
  1. Identify the emotional state in which the all the quoted dialogues
     should be spoken based on the context, identify the intensifying and
     contrasting emotions.
  2. Provide precise timestamps of **EVERY** crucial part of the quoted
     dialogue, and comment on the emotional expressiveness of these parts
     that are imporant to convey the **OVERALL** emotional tone of the
     dialogue.
  3. Analyze the boundry points, where quoted dialogues and narrative
     context meet, and provide precise timestamps of these parts, while
     reasoning how there may be a change in the emotional tone of the
     speech at these points.
  4. Finally, reason deeply and justify how the expressive the TTS system
     is, and how it has narrated the **text_to_synthesize**, and then what
     the final score for the TTS system should be.
""" ,

"Syntactic Complexity" : """
**Evaluation Dimension:**
  - In this category, we want to evaluate the ability of the TTS system to
    use **prosody (pausing, phrasing, intonation, stress)** to make
    complex sentence structures easily understandable.
  - It tests if the TTS can convey a syntactically very complex sentence
    such that it's meaning to the listener is clear and understandable,
    that is the main goal.
  - Occasionaly, the text may contain homographic words, in that case, the
    TTS system should pronounce the homographic words with appropriate
    pronunciation.

**Example:** "The book that the professor who won the award wrote is on the
  table."
**Explanation:**
  - Without proper phrasing and intonation, it's hard to follow who did
    what or to identify the main subject ("the book") and the verb ("is")
    .
  - The rest of the sentence-"that the professor who won the award wrote"-
    is a complex noun modifier (a series of nested relative clauses)
    describing "the book."
  - The core structure of the sentence is: "The book is on the table."
  - A TTS system must use appropriate prosody-pausing, stress, and
    intonation-to guide the listener naturally through the structure,
    signaling the main subject, distinguishing the embedded clauses, and
    connecting all parts coherently.

**Note:**
  - This category is all about adding appropriate pauses, stress, and
    intonation, in absence of punctuation marks, **AND** in their
    presence too. We want to check if the indended meaning is conveyed
    correctly and that is all that matters.

**Rating Scale:**
1: The prosody makes the sentence structure confusing or leads to an
   incorrect meaning.
2: The intended structure is mostly understandable, but the prosody (pauses,
    intonation, stress) sounds unnatural or confusing at some parts.
3: The prosody correctly conveys the sentence structure, making the complex
   grammar easy to follow and clarifying the intended meaning of the
   sentence very clearly.

**Reasoning guidelines:**
1. Elaborate the intended meaning of the sentence and untangle the complex
   syntax.
2. Identify the syntactically complex parts of the speech that require
   appropriate prosody (pausing, phrasing, intonation, stress) to be

understandable, also identify any homographs and their intended
       pronunciation, finally list all these crucial parts.
    3. Carefully analyze and provide precise timestamps of crucial prosodic
       features - pauses between phrases, changes in intonation, and stress
       patterns - that help clarify the sentence structure for each of the
       crucial parts.
    4. Evaluate how well the prosody helps to distinguish the meaning at these
       crucial parts, for example, distinguish between main clauses and
       subordinate clauses, avoid garden path effects, and other syntactic
       complexities(including homographs) identified in 2.
    5. Finally, reason deeply and justify how effectively the TTS system's
       prosodic features (or lack thereof) contribute to the comprehensibility
       of the **OVERALL**complex syntax, and then determine the final score for
        the TTS system.
    """ ,

    "Foreign Words" : """
    **Evaluation Dimension:**
        - In this category, we want to evaluate the ability of the TTS system to
            correctly pronounce foreign words and phrases, either using their
            original pronunciation or a widely accepted anglicized version.
        - The goal for the system is to sound like a fluent bi-lingual speaker,
            seamlessly doing code-switching between the languages.

    **Example:** "During his shaadi, manoj went pura paagal and started dancing
        jaise ki wo ek actor hai."
    **Explanation:**
        - The words "shaadi", "paagal", "jaise" and "actor" should be pronounced
            with an acceptable hindi pronunciation(as there is no anglicized
            version for these words).
        - The flow when switching between the two languages should be seamless
            and natural, without awkward pauses or jumps.

    **Note:**
        - Not all foreign words have an anglicized version, in that case the
            words should be pronounced with an acceptable pronunciation in that
            foreign language.

    **Rating Scale:**
    1: Pronounces the foreign words and phrases completely incorrectly.
    2: Applies foreign pronunciation but not entirely correctly, some words are
        pronounced correct but others are not and the natural flow during code-
        switching is disrupted.
    3: Correct rendering in the intended language or acceptable anglicized
        version for all words and phrases, and the natural flow during code-
        switching is maintained.

    **Reasoning guidelines:**
        1. Identify the foreign words and phrases, and the language they belong
            to.
        2. Provide precise timestamps for **ALL** the foreign words and phrases
            in the speech.
        3. Analyze the audio multiple times, and provide a comment on the
            pronunciation of the foreign words, and if the system has gotten none
            , some or all of them correct.
        4. Finally, reason deeply and justify how the TTS system has performed
            based on pronunciation **AND** the flow at code-switching points, and
             then what the final score for the TTS system should be.
    """ ,

    "Paralinguistics" : """
    **Evaluation Dimension:**
        - In this category, we want to evaluate how well the TTS system synthesis
             speech corresponding to paralinguistic cues present in the text.

There can be multiple types of paralinguistic cues present in the
text, like:
- Interjections ("Hmmm", "Ooops").
- Vocal sounds/onomatopoeia("Shhh!", "Achoo!", "Meow")
- Emphasis using CAPS("He didn't REALLY mean it" has a different
    sound than "He didn't really MEAN it"), vowel elongation("
    Heyyyyyyy, okayyyyyyy"), hyphenation/syllable stress("ab-so-
    lutely", "im-por-tant"), etc.
- Pacing cues(ellipses, punctuation(for example STOP.RIGHT.THERE)).
- Stuttering and hesitation("I-I-I", "W-we-well...", etc.)
- The TTS system has to correctly identify all the paralinguistic cues
    present in the text and render them how human speech would render
    them.

**Example:** '"Ugh! I-I told you... DO NOT touch that! Seriously?!"'
**Explanation:** The TTS should render the frustration ("Ugh!"), hesitation
    ("I-I", "..."), emphasis ("DO NOT"), and final incredulous annoyance ("
    Seriously?!") suggested by the text, not just read the words flatly.

**Note:**
- It is **VERY IMPORATANT** to recognize that we are looking for a
    plausible rendering of the paralinguistic cue, as a human would
    render them while speaking the text.
- Paralinguistic realism is also affected by the emotional tone that cue
    represents, you will only focus on the emotional affect for the cues,
    not the emotional tone forother parts of the speech.

**Rating Scale:**
1: Fails to render the intended vocal effect(s); sounds neutral or wrong.
2: Intention to render the vocal effect(s), but the delivery sounds
    unnatural, awkward, or inaccurate.
3: Successfully and naturally produces the vocal effect(s) implied by the
    textual cues.

**Reasoning guidelines:**
   1. Identify and list all of the paralinguistic cues present in the text,
       and the plausible intended vocal effect for each of them.
   2. Provide precise timestamps for **ALL** the paralinguistic cues in the
       speech.
   3. Give detailed analysis for **ALL** cues by analyzing the audio
       multiple times, like how they are synthesized, if they match the
       intended vocal effect, and how realistic to human speech they are.
   4. Finally, reason deeply and justify how the TTS system has performed in
       rendering the paralinguistic cues, and then what the final score for
       the TTS system should be.
""" ,

"Pronunciation" : """
**Evaluation Dimension:**
   - In this category, we want to evaluate how well the TTS system
       pronounces non-trival words, numerals and special characters present
       in the text.
   - To be specific, this category includes **text_to_synthesize** that fits
       in ONE of the following complex pronunciation categories and the TTS
       system has to render the **text_to_synthesize** correctly:
      1. Text with currency and numerals in different formats
      2. Text with dates and time-stamps in different formats
      3. Texts with email addressess, passwords and urls in different
          formats.
      4. Texts with complex street addresses or location references.
      5. Texts with equations and notations from the STEM field.
      6. Texts that have **BOTH** an initialism(pronounced initial by
          initial) and acronym(pronounced as a whole word).
      7. Texts with repeated tounge twisters.

```
        **Example:** "The equation e^(i*pi) + 1 = 0 is a famous equation in
            mathematics."
        **Explanation:**
            - The equation "e^(i*pi) + 1 = 0" should be pronounced with the
                appropriate pronunciation, like "The equation e to the power of i
                times pi plus 1 equals 0 is a famous equation in mathematics."

        **Note:**
            - It is crucial to understand what the most natural pronunciation of the
                given text will be, sometimes it maybe helpful to think in reverse, i
                .e, if **text_to_synthesize** is actually the transcription of an
                audio, what would that audio sound like? The TTS system should
                synthesize audio similar to that.
            - It is more ideal for a system to speak tounge twisters faster **WHILE**
                still maintaining complete clarity **AND** consistency in
                pronunciation.
            - Initialisms should be pronounced initially by initial(for example, FBI)
                , and acronyms(for example, NASA) should be pronounced as a single
                word.
            - Case-sensitivity sometimes matters(for example, passwords, URL paths
                after the domain name, etc.), so make sure to recognize any case-
                sensitive parts and reward/penalize accordingly.

        **Rating Scale:**
        1: Incorrect synthesis of the critical parts, with missing or completely
            incorrect/inappropriate pronunciation.
        2. Partially correct pronunciation of the **SOME** of the critical parts.
        3. Completely correct pronunciation of **ALL** the critical parts.

        **Reasoning guidelines:**
            1. Identify the critical parts of the text that require correct
                pronunciation.
            2. Provide precise timestamps for **ALL** the critical parts in the
                speech, and the ideal pronunciation for the same.
            3. Give detailed analysis for **ALL** the critical parts by analyzing the
                audio multiple times, explain how they are synthesized, if they
                match the intended pronunciation, and how realistic to human speech
                they are.
            4. Finally, reason deeply and justify how the TTS system has performed in
                pronouncing the critical parts, and then what the final score for
                the TTS system should be.
        """
    }
```

# D   Analysis of Gemini-2.5-Pro as a Judger and the case of Audio Subjectivity

In this section, we analyze Gemini-2.5-Pro's behavior as a judge across categories, examining both its strengths in detecting nuanced differences and its limitations in subjective scenarios.

**Questions:**   Gemini demonstrates strong capability in recognizing intonation patterns, correctly identifying rising and falling contours in most cases. The tie-breaking procedure works effectively, with the judge appropriately preferring subtle prosodic advantages (e.g., choosing natural rising intonation over flat delivery when both systems score equally). However, occasional misclassifications occur where flat intonation is incorrectly perceived as rising/falling, or where tie-breaking is applied to equivalent performances. These edge cases often involve subjective interpretations of how preceding words contribute to overall interrogative prosody beyond just the final pre-question-mark intonation.

**Emotions:**   While Gemini consistently identifies intended emotions from textual context and reliably rewards systems with perceptible emotional variation (e.g., GPT-4o-audio Ballad vs. baseline), challenges emerge with emotionally flat outputs. In such cases, the judge occasionally hallucinates emotional expression where none exists. Additionally, as noted in Section 4.4, voice characteristics

introduce systematic biases: high-pitched voices may advantage certain emotions while deeper voices favor others. Close comparisons in this inherently subjective category often depend on subtle interpretative judgments.

**Foreign Words:** Gemini excels at phonemic analysis, providing evidence-based reasoning by correctly matching synthesized sounds to intended pronunciations. For clear cases (heavily anglicized vs. native pronunciation), performance is robust. Remarkably, the judge detects subtle phonemic distinctions, such as correctly identifying when Spanish "tocayo" is pronounced "toh-KAI-yoh" instead of "toh-KAH-yoh"-differences sometimes requiring multiple human listening passes. However, this sensitivity sometimes leads to over-emphasis of minor phonetic variations, resulting in tie-breaking or scoring differences for similar pronunciations.

**Paralinguistics:** The judge shows comprehensive understanding across all paralinguistic cues-interjections, onomatopoeia, emphasis markers, pacing cues, and stuttering. It accurately maps textual cues to vocal sounds, recognizing elongation, syllable stress, and capitalization emphasis. Fine-grained distinctions are captured, such as rewarding crisp "Pssst" rendering over less precise vocalizations. However, subjectivity in duration judgments (e.g., optimal length for "heyyyyyy") occasionally produces winner selection based on minimal temporal differences. Complex hyphenated emphasis like "TRU-ly ter-RI-ble" is handled well, by penalizing a system that does strict word-splitting errors while rewarding pronunciation that ignores hyphenation cues but still produces a natural pronunciation.

**Syntactic Complexity:** Gemini reliably focuses on pausing and stress patterns crucial for syntactic disambiguation. Homograph resolution is particularly strong-correctly identifying when ElevenLabs rendered "minute-by-minute" with inconsistent pronunciations (my-NOOT and min-it) when both should be "min-it." Occasional errors involve misperceiving pause durations, either over- or under-estimating their length in the audio.

**Complex Pronunciation:** This category exhibits negligible hallucinations, leveraging Gemini's robust ASR capabilities. The judge provides detailed reasoning about which components are synthesized more accurately, enabling precise and fine-grained winner determination based on granular pronunciation analysis.

**Subjectivity and Human Agreement:** Our human evaluation study yields a Krippendorff's $\alpha$ of $0.5073$, indicating weak-to-moderate inter-annotator agreement and confirming the subjective nature of many TTS quality judgments. This weak agreement reflects the genuine difficulty humans face in consistently evaluating expressive speech synthesis.

**Implications for Automated Evaluation:** Despite any observed limitations, Gemini-2.5-Pro's biases and occasional hallucinations are outweighed by crucial advantages for large-scale evaluation. Unlike human judges, the model provides consistent, reproducible assessments across thousands of samples, detailed timestamp-based reasoning, and scalable evaluation at a fraction of human annotation costs. The high correlation with human preferences ($90\%+$ Spearman correlation) and strong inter-judge agreement across different LALMs (Kendall's $W = 0.97$) demonstrate that while individual judgments may not be perfect, the overall ranking and comparative analysis remain reliable and actionable for TTS system development.

# E   Text Normalization prompt and examples

Detailed below is the prompt we use when GPT-4.1-mini acts as the Text Normalizer for the results present in Table 3a. We use

```
normalize_prompt = """
Normalize the following text for text-to-speech processing. Convert numbers, symbols
    , units, formulas, addresses, URLs, email addresses, dates, times, currencies,
    measurements, scientific notation, and abbreviations into their spoken form.
    Maintain natural reading flow by handling punctuation appropriately.

For numbers:
```

```
- Expand decimal numbers (e.g., "3.14" -> "three point one four")
- Express fractions as "X over Y" or use ordinals for common fractions
- Write percentages as "X percent"
- Handle currency symbols and values naturally
- Treat phone numbers, postal codes, and IDs as individual digits

For symbols and special characters:
- Explain mathematical symbols (+/-, x, etc.)
- Express chemical formulas appropriately
- Describe non-ASCII characters clearly

For abbreviations and acronyms:
- Read common acronyms as words if pronounceable (NASA, UNESCO)
- Spell out other acronyms letter by letter (FBI as "F B I")
- Expand standard abbreviations (St. -> Street, Dr. -> Doctor)

For specialized content:
- Read URLs and email addresses component by component
- Express time zones and scientific units naturally
- Handle coordinates, addresses, and references in a clear, conversational manner

You will output only the normalized text, which will be sent directly as input to
    the TTS model, do not output any other additional information or text.
Text to normalize:
{{{text_to_synthesize}}}
"""
```

In addition to the cases we mention in Section 4.5, we present some additional cases observing the effect of WeText-TN [4] and GPT-4.1-mini-TN.

**WeText-TN:** `Worked Correctly:` (i) $2\frac{1}{3}\%$ → *two and one third percent*; (ii) $v4.0.1170 - rc.3b+$ → *version v four point oh point one one seven oh rc point three b+*; (iii) 2024-09-1 → *fifteenth of September twenty twenty four.*
`Worked Incorrectly:` (i) Ste. 1250-B → Did not expand to *Suite*; (ii) $\sim \$1,670.83$ → *tilde* instead of *approx*; (iii) $10^3$ mL → *Ten ³* instead of *Ten power 3*; (iv) 12/19/24-01/12/25 → *twelve divided by nineteen divided by twenty...* instead of reading as a *date range*; (v) CRISPR-Cas9 → *CRISPR-CA's nine* instead of pronouncing as word.

**GPT-4.1-mini TN:** `Worked Correctly:` There are multiple cases across numerals, currencies, passwords, web-addresses, etc that worked very well with this TN technique.
`Worked Incorrectly:` (i) $1.075005$ → *one thousand seventy-five point zero zero five* instead of *one point zero seven five zero zero five*; (ii) UTC+11:00 → *Coordinated Universal Time plus 11 hours* **not** preferred by judger over *U T C plus eleven hours*; (iii) $\$1,890.125375$ → *... dollars and twelve cents five three seven five*, this is misleading way to represent currency; (iv) $\$12.40\frac{1}{2}$ → $\$12.40$ 5/10, a case of *over-normalization*; (v) Many cases where abbreviations supposed to pronounced as a single word are separated letter by letter, and cases where abbreviations being expanded is not preferred by the judger.

