# OpenReview forum: "EmergentTTS-Eval: Evaluating TTS Models on Complex Prosodic, Expressiveness, and Linguistic Challenges Using Model-as-a-Judge"
_NeurIPS.cc/2025/Datasets_and_Benchmarks_Track — NeurIPS 2025 Datasets and Benchmarks Track poster_

### Official Review · Reviewer_v3gf · 2025-06-23

**Rating:** 5
**Confidence:** 5

**Summary:**

This work proposes a new benchmark to evaluate TTS systems comprehensively. First, an approach to curating test cases is proposed, which utilizes LLMs to change normal texts into complex texts in six scenarios. Then, an approach based on LALMs is proposed to evaluate the performance of a TTS system on this large test set. Finally, the work presents the performance of various TTS systems on this benchmark, demonstrating the effectiveness of this new benchmark in evaluating TTS.

**Dataset Code Accessibility:**

Yes

**Ethical Considerations:**

No, there are no or only very minor ethics concerns

**Final Justification:**

After discussing with the author, I decided to keep the rating. The novelty of this paper is good. All concerns are all addressed in the author's response. Some confusing points will be clarified in the updated version. Hence, I will recommand accepting this paper.

**Limitations Weaknesses:**

1. Not well discussing the relationship between win-rate and MOS. MOS has been the most important metric for TTS evaluation. But the proposed win-rate does not align with MOS. This significant difference worth a deeper discussion. Otherwise, the win-rate may be not convincing.
2. How is MOS test conducted? Details should be presented.

**Strengths Contributions:**

1. Propose a novel approach to create test sets by using LLMs. This approach indeed can curate many complex texts that is hard to design by hand only.

2. Propose a new automatic evaluation approach based on LALMs.

---

> ### Author Rebuttal · Authors · 2025-07-31
>
> Thank you for the positive feedback! Here are our responses to the questions:
>
> **Q1. Not well discussing the relationship between win-rate and MOS. MOS has been ... the proposed win-rate does not align with MOS. This significant difference worth a deeper discussion. Otherwise, the win-rate may be not convincing.**
>
> We evaluated additional models after our initial submission and in response to reviewers feedback. Below, we present the updated correlation analysis between win-rates and MOS scores, showing Spearman correlations for overall performance and individual categories:
>
> | Category               | Spearman Correlation (Win-Rate vs. MOS) |
> | :--------------------- | :-------------------------------------- |
> | **Overall**            | **30.77%**                          |
> | Pronunciation          | 42.43%                                  |
> | Questions              | 41.63%                                  |
> | Syntactic Complexity   | 31.54%                                  |
> | Foreign Words          | 22.74%                                  |
> | Emotions               | 20.77%                                  |
> | Paralinguistics        | 9.31%                                  |
>
> We also refer to the table presented for reviewer 2VbD that shows "Gemini 2.5 Pro win rate" has much higher correlation with human preference than wv-mos.
>
> | Metric               | Spearman Correlation with Human Preference |
> |-------------------|----------------|
> | Gemini 2.5 Pro win rate   | **90.48%**                |
> | wv-mos [1]                    | 21.43%                |
>
> Here is a deeper analysis of why the divergence between MOS and win-rate exists.
>
> MOS scores were originally designed to evaluate overall audio quality and naturalness, focusing primarily on technical aspects like clarity and intelligibility. However, our benchmark evaluates more nuanced semantic, expressive and linguistic capabilities that MOS does not focus on specifically. For categories like Emotions and Paralinguistics, which require understanding of subtle vocal expressions and contextual appropriateness, MOS shows particularly weak correlation (20.77% and 9.31% respectively) because these aspects fall outside its design scope. At the same time, a TTS model producing unintelligble numbers and emails in the pronunciation category will have a low MOS and low win-rate. Additionally, the correlation patterns may vary depending on the specific MOS prediction model architecture and training data used.
>
> Our win-rate, directly evaluates whether the TTS output correctly conveys the intended semantic meaning and emotional expression based on specific test-cases. This aligns much better with human preference (90.48% correlation) because it captures the nuanced understanding that humans naturally apply when judging TTS quality in specific contexts. We believe that MOS and win-rate should be viewed in complement of each other: MOS remains valuable for technical quality assessment, while our win-rate fills the critical gaps of evaluating specific capabilities of TTS models.
>
> **Q2: How is MOS test conducted? Details should be presented.**
>
> We use an automatic MOS prediction model called wv-mos, as mentioned in Section 4.1 of the paper (line 218). For the human correlation survey, the details are presented in Section 4.6 of the paper.
>
> [1] Andreev, Pavel, et al. "HIFI++: A Unified Framework for Bandwidth Extension and Speech Enhancement". In ICASSP 2023 - 2023 IEEE International Conference on Acoustics, Speech and Signal Processing (2023)

---

> > ### Comment · Reviewer_v3gf · 2025-08-04
> >
> > Thanks for your response. I would like to give my opinions on the relationship between win-rate and MOS as a reference.
> >
> > MOS can be used to evaluate all aspects of the audio if you can hire sufficient listeners and guide them on how to evaluate it. AutoMOS, which can be seen as a kind of model-as-judge, leverages existing subjective scores to achieve automatic evaluation. Hence, it is crucial to collect abundant MOS tests on "Emotions", and then use the data to train AutoMOS. It lacks efficiency and flexibility. That is why your approach presents greater potential.
> >
> > Regarding Q1, you were not comparing with MOS, but an AutoMOS model. MOS test is always effective if you can conduct it well, but an autoMOS model is still defective. You need to clarify this.
> >
> > Regarding Q2, you need to clarify what the AutoMOS model is evaluating by investigating its training set.

---

> > ### Author Response · Authors · 2025-08-05
> >
> > Thank you for recognizing that our proposed approach offers greater potential than automated MOS scoring! Traditional MOS prediction models are typically trained on domain-specific listening test datasets, making them difficult to adapt to new evaluation criteria (e.g., emotional expressiveness). The VoiceMOS Challenge [1], a key evaluation benchmark for MOS prediction models, focuses on general naturalness and signal quality, without asking listeners to judge specific prosodic or expressive attributes for curating their datasets. In contrast, our model-as-a-judge framework leverages a general-domain audio understanding model, offering greater flexibility and efficiency for diverse assessment tasks. We have presented evidence for robust capabilities of Gemini 2.5 pro as judge in response to reviewer 2VbD.
> >
> > Following your suggestion, we will clarify that the currently cited wv-mos model [2] based on the wav2vec 2.0 architecture is an automated MOS prediction system and was trained on listening test results from the Voice Conversion Challenge (VCC) 2018, which reflect perceived audio naturalness. This accounts for its divergence with the win-rate and our human-preference study in Section 4.6 as both focus on the specific aspects of speech.
> >
> > [1] Huang, Wen-Chin, et al. "The VoiceMOS Challenge 2024: Beyond Speech Quality Prediction", arXiv preprint arXiv:2409.07001 (2024).
> >
> > [2] Andreev, Pavel, et al. "HIFI++: A Unified Framework for Bandwidth Extension and Speech Enhancement". In ICASSP 2023 - 2023 IEEE International Conference on Acoustics, Speech and Signal Processing (2023).

---

### Official Review · Reviewer_2VbD · 2025-07-03

**Rating:** 5
**Confidence:** 3

**Summary:**

The authors introduce EmergentTTS-Eval, a comprehensive, extensible benchmark that covers six complex TTS scenarios and leverages LLMs for evaluation. Using a Large Audio Language Model (LALM) as an automated judge, the framework assesses diverse TTS outputs across multiple dimensions, revealing differences between systems with strong correlation to human judgment.

**Dataset Code Accessibility:**

Yes

**Dataset Code Comments:**

The dataset and code can be access by github and huggingface.

**Ethical Considerations:**

No, there are no or only very minor ethics concerns

**Final Justification:**

I appreciate the authors’ responses. After carefully reviewing the rebuttal, I think most of the issues have been addressed, and I increased my score.

**Limitations Weaknesses:**

- Regarding the use of LLM-based judges, although the authors compare several different LLM judges, I am curious about how they evaluate the judges’ ability to perceive emotional and paralinguistic information. For example, can Gemini 2.5 Pro or other LLM judges accurately identify emotions in speech without being biased by the textual content? Similarly, can these models accurately detect prosodic features such as emphasis? A thorough analysis on this would be very helpful. If the LLM judges lack the ability to do so, the reliability of the evaluation results may be significantly compromised.

- In addition, it lacks results for many popular open-source models, such as MaskGCT, CosyVoice2, E2 TTS, and F5 TTS.

**Strengths Contributions:**

- The paper is easy to understand, and the experimental details are clearly presented.

- The work evaluates TTS systems from several novel perspectives.

---

> ### Author Rebuttal · Authors · 2025-07-31
>
> We thank the reviewer for their valuable feedback and for finding our work easy to understand. We are especially encouraged that the reviewer recognized our core goal: to evaluate TTS systems from the novel and challenging perspectives that current benchmarks often miss. Following are the responses to their questions.
>
> **Q1: Curious about...the judge's ability to perceive emotional and paralinguistic information...can these models accurately detect prosodic features such as emphasis? A thorough analysis...**
>
> Thanks for the insightful question! The LALM's ability to comprehend audio is crucial to its effectiveness as a judge. Below, we present both qualitative and quantitative evidence demonstrating that our chosen judge, Gemini 2.5 Pro, is a strong model for audio understanding and TTS evaluation.
>
> ### Qualitative Evidence
>
> In Appendix E of the submission, we provided an in-depth explanation and a few case-studies of Gemini 2.5 Pro as a judge for each category in our benchmark. In summary, we found that the model excels at: (1) recognizing intonation patterns, (2) identifying intended emotions from textual context and reliably rewarding perceptible emotional variation, (3) offering evidence-based reasoning by correctly matching synthesized audio to intended pronunciations of foreign words, (4) understanding paralinguistic cues, 5) detecting pausing and stress patterns crucial for syntactic disambiguation, and (6) transcribing audio accurately even when it includes complex pronunciations.
>
> ### Quantitative Evidence
>
> While our work's novelty means no prior benchmark exists for this exact LALM-as-a-judge task, we can validate our choice of judge, Gemini 2.5 Pro, through some surrogate metrics, including on LALM understanding benchmarks from recent concurrent work [2, 3, 4].
>
> 1. **Correlation with Human Preference**: Gemini 2.5 Pro's judgement has high alignment with human rankings and achieved spearman correlation of 90.48%, which is also statistically significant (p-value 0.002). As shown in the table below, Gemini's win-rate has a much higher correlation with human preference when compared with the wv-mos [1] MOS prediction model we used.
>
>     | Metric               | Spearman Correlation with Human Preference |
>     |-------------------|----------------|
>     | Gemini 2.5 Pro win rate   | **90.48%**                |
>     | wv-mos                    | 21.43%                |
>
> 2. **Emotion Recognition Accuracy**: While the reviewer raised valid concerns about textual bias, our benchmark mitigates this by directly comparing two audio samples for emotional expressiveness rather than relying on text-based evaluations. Additionally, we draw support from a concurrent research that proposed the EmoNet-Voice [2] benchmark. This paper assessed models on nuanced emotion recognition across 40 distinct emotional intensity labels. According to the paper, Gemini 2.5 Pro is the top-performing general-purpose model in terms of correlation with expert human annotations (corr=0.417). This is on par with their own model trained specifically on an emotion recognition dataset (corr=0.421). Notably, this level of correlation is significant given the low inter-rater agreement among human annotators themselves (Cronbach's $\\alpha$=0.14). In Appendix E of the paper, we also reported that the inter-annotator agreement is weak-to-moderate (Krippendorff's $\\alpha$=0.5073). The result shows that it is challenging for human to give consistent scores to speech expressiveness. Our approach offers a more reproducible and scalable alternative to large-scale human evaluation for assessing a TTS model's ability to generate emotional speech.
>
> 3. **Prosody (Emphasis/Stress) Detection**: We adapted StressTest [3] into a pairwise comparison task. The model must identify which of two audio samples correctly emphasizes a specified word from the text, a setup that directly mirrors our benchmark's methodology. Our results show that Gemini 2.5 Pro achieves an accuracy of 95.11% on this task, significantly outperforming the 50% baseline of random guessing. The result is statistically significant, with a 95% confidence interval of ±2.59%. This result demonstrates that Gemini 2.5 Pro is highly reliable in evaluating emphasis.
> 4. **Performance on Step-Audio-2 Paralinguistic Benchmark**: We run Gemini 2.5 Pro on the paralinguistic benchmark proposed by Step-Audio-2 [4]. We selected six categories from this benchmark that directly probe the core components of prosodic and emotional perception: `Emotion`,`Pitch`, `Rhythm`, `Speed`, speaking `Style` (e.g., distinguishing colloquial vs. narrative speech), and non-speech `Vocal` effects (e.g., identifying laughter or sighs).
>
>     The benchmark measure the accuracy of detection and results show that our chosen judge, Gemini 2.5 Pro, demonstrates state-of-the-art performance among general-purpose audio models, achieving the highest average score across these crucial dimensions and performing competitively even against the benchmark's own specialized model.
>
>     | Model                       | Paralinguistic Avg. | Emotion | Pitch | Rhythm | Speed | Style | Vocal |
>     | --------------------------- | ------------------- | ------- | ----- | ------ | ----- | ----- | ----- |
>     | Step-Audio-2 (Benchmark Model) | 77.33               | 72      | 78    | 70     | 78    | **84**    | 82    |
>     | **Gemini 2.5 Pro**          | **82.00**           | **86**      | **84**    | **74**     | **84**    | 80    | **84**    |
>
>    These results confirm that Gemini 2.5 Pro possesses state-of-the-art capability to understand the various prosodic and emotional features-from pitch contours to specific vocal effects-that our benchmark cares about.
>
> **Q2: In addition, it lacks results for many popular open-source models, such as MaskGCT, CosyVoice2, E2 TTS, and F5 TTS.**
>
> We thank the reviewer for their concern. Since the paper submission, we evaluated another five popular open-source models: ChatterBox [5], VITS [6], F5-TTS [7], Kokoro-82M [8], and KyutAI-TTS [9]. The table below summarizes the win-rate of additional models evaluated:
>
> | Model                        | Overall | Emotions | Foreign Words | Paralinguistics | Complex Pronunciation | Questions | Syntactic Complexity |
> |------------------------------|---------------------|----------------------|---------------------------|------------------------------|----------------------------------|----------------------|----------------------------------|
> | ResembleAI Chatterbox        | 26.74%            | 22.48%              | 26.42%                   | 23.75%                     | 8.36%                            | 48.57%               | 28.57%                         |
> | Kokoro-82M                   | 23.46%            | 13.92%              | 12.67%                   | 10.00%                     | 23.06%                            | 40.89%               | 40.17%                         |
> | KyutAI-TTS                   | 20.91%            | 29.46%              | 16.07%                   | 26.96%                     | 6.73%                            | 30.00%               | 14.46%                         |
> | F5TTS                        | 15.31%            | 26.78%              | 1.78%                    | 21.60%                     | 1.42%                            | 14.82%               | 23.75%                         |
> | VITS-VCTK                    | 7.64%             | 0.00%               | 4.54%                    | 4.10%                      | 17.82%                            | 15.53%               | 5.07%                         |
>
>
> References:
>
> [1] Andreev, Pavel, et al. "HIFI++: A Unified Framework for Bandwidth Extension and Speech Enhancement". In ICASSP 2023 - 2023 IEEE International Conference on Acoustics, Speech and Signal Processing (2023)
>
> [2] Schuhmann, Christoph, et al. "EmoNet-Voice: A Fine-Grained, Expert-Verified Benchmark for Speech Emotion Detection" arXiv preprint arXiv:2506.09827 (2025).
>
> [3] Yosha, Iddo, et al. "StressTest: Can YOUR Speech LM Handle the Stress?" arXiv preprint arXiv:2505.22765 (2025).
>
> [4] Wu, Boyong, et al. "Step-Audio 2 Technical Report" arXiv preprint arXiv:2507.16632 (2025).
>
> [5] HuggingFace: "ResembleAI/chatterbox"
>
> [6] Kim, Jaehyeon, et al. "Conditional Variational Autoencoder with Adversarial Learning for End-to-End Text-to-Speech" arXiv preprint arXiv:2106.06103 (2021).
>
> [7] Chen, Yushen, et al. "F5-TTS: A Fairytaler that Fakes Fluent and Faithful Speech with Flow Matching" arXiv preprint arXiv:2410.06885 (2025).
>
> [8] HuggingFace: "hexgrad/Kokoro-82M".
>
> [9] HuggingFace: "kyutai/tts-1.6b-en_fr".

---

> > ### Comment · Reviewer_2VbD · 2025-08-06
> >
> > I appreciate the authors’ responses. After carefully reviewing the rebuttal, I think most of the issues have been addressed, and I increased my score.

---

### Official Review · Reviewer_71y1 · 2025-07-03

**Rating:** 4
**Confidence:** 3

**Summary:**

This paper proposes an LLM-based evaluation method as an alternative to human evaluation-based MOS for assessing the performance of text-to-speech (TTS) models.

1. It defines 6 challenging tasks for TTS models (emotions, paralinguistics, etc.).

2. For each task, a small set of evaluation texts is manually written, and then expanded into more difficult sentences using an LLM, resulting in a total of 1,645 evaluation texts.

3. The evaluation texts are synthesized into speech using TTS models.

4. For each pair of TTS outputs, a large audio language model (Gemini 2.5 Pro) selects the better one. A win-rate-based metric is then used to determine model performance, where a higher Win-Rate indicates better quality.

**Additional Feedback:**

This paper presents a simple yet effective evaluation method (6 tasks and Win-Rate) along with a dataset. However, the proposed evaluation may only be valid for large-scale TTS models, and could result in unfair assessments for most typical TTS models. To increase the contribution of this work, the authors should demonstrate through experiments that their evaluation method is also applicable to TTS models trained on standard datasets. Additionally, it would strengthen the paper if the authors clearly explain why the proposed approach is more effective compared to commonly used objective metrics in TTS evaluation.

**Dataset Code Accessibility:**

Yes

**Dataset Code Comments:**

All code and dataset links provided in the paper are fully functional and come with clear and thorough documentation.

**Ethical Considerations:**

No, there are no or only very minor ethics concerns

**Final Justification:**

Through the rebuttal, the authors have addressed my two main concerns:
first, whether the proposed method can be applied to small-scale TTS models, and second, its correlation with objective metrics.
The rebuttals for both points appear reasonable. Therefore, I am raising my score from 3 to 4.

**Limitations Weaknesses:**

For 6 challenging tasks proposed in this paper, valid evaluation is likely only possible on TTS models trained with extremely large-scale datasets (e.g., GPT-4o-mini-TTS, Qwen 2.5 Omni, etc.). This is because TTS models trained on typical datasets (such as VCTK, LibriTTS, Multilingual LibriSpeech, etc.) are incapable of handling some tasks involving emotions, paralinguistics, and complex pronunciation. Therefore, the scope of TTS models to which the evaluation tasks proposed in this paper can be applied is highly limited.

Furthermore, it would be beneficial if the paper could clarify how the Win-Rate metric used in this study correlates not only with MOS but also with objective measures such as word error rate (WER) and PESQ.

**Strengths Contributions:**

This paper proposes a novel evaluation metric for text-to-speech (TTS) models, which involves generating speech from challenging texts and measuring the win rate of each TTS model.

1. To simulate real-world application scenarios, the authors generate challenging evaluation texts using large language models (LLMs).

2. The most widely used metric for evaluating TTS quality is human evaluation-based MOS (Mean Opinion Score). However, MOS requires significant time and effort due to the need for manual evaluations. To address this issue, the paper introduces a LALM-based (Large Audio Language Model) evaluation framework, enabling a fully automated assessment process.

3. The proposed LALM-based Win-Rate metric correlates well with MOS. Experimental results show that TTS models with high MOS scores also achieve high Win-Rates under the LALM-based evaluation.

---

> ### Author Rebuttal · Authors · 2025-07-31
>
> We thank the reviewer for their valuable feedback and for recognizing the key contributions of our work. We are particularly grateful for their positive assessment of our novel win-rate metric, our methodology for generating challenging real-world evaluation texts, and our framework's value as a fully automated alternative to time-consuming human evaluations. We address the reviewer's concerns regarding the benchmark's applicability and its correlation with objective metrics below.
>
> **Q1: The proposed evaluation may only be valid for large-scale TTS models, and could result in unfair assessments for most typical TTS models.**
>
> We want to highlight that smaller TTS models can be effectively evaluated with our framework and do not result in "unfair" assessment. We evaluate a large number of open-source models that are trained with smaller-scale data or have lower parameter count and can see clear difference in their win-rates. For example, Sesame1B [1] achieves 15.96% overall win-rate while Orpheus TTS [2] achieves 30.12% win-rate. Furthermore, the category-wise win-rate acts as a solid diagnostic tool for these models. For example, we can see from Table 1 that although QWen 2.5 Omni [3] is better than "Orpheus TTS", "DeepGram Aura-2" and "11Labs eleven multilingual v2" in "Emotions", it is worse than these models in handling paralinguistics.
>
> During the rebuttal period, we evaluated additional open-source models (VITS [4], Kokoro-82M [5], KyutAI [6] and F5-TTS [7]) to further support this point and their detailed results can be found in our response to reviewer 2VbD. Kokoro-82M, trained on only a few hundred hours of audio, achieves a 23.06% win rate in the "complex pronunciation" category - outperforming all open-source TTS models, which typically fall below a 10% win rate. Moreover, Kokoro-82M reaches up to 40% win rate in the "questions" and "syntactic complexity" categories. These results suggest that its phoneme-based modeling approach is particularly effective for handling difficult-to-pronounce words, complex sentence structures, and different types of questions. However, the model performs less well in the "emotions" category, with a win rate of just 13%, indicating a limitation in expressive narration. VITS, a model trained on the 44-hour VCTK [8] dataset, achieves a win-rate of 17% on complex pronunciation, higher than all other open-source models we evaluated and only behind Kokoro-82M. Finally, F5-TTS, trained primarily on the open-source Emilia [9] dataset, achieves a win-rate of 27% in the "Emotions" category, putting it at-par with large-scale systems like "Deepgram-Aura-2" and "11Labs eleven multilingual v2".
>
> Our framework provides unique value by enabling direct comparative analysis between any TTS models across different categories and complexity levels. While our benchmark uses state-of-the-art models like GPT-4o-mini-TTS as reference points for win-rate calculations, the approach still reveals meaningful performance differences among smaller models. To the best of our knowledge, EmergentTTS-Eval is the first benchmark that offers such granular comparative evaluation. We are confident that our framework helps users identify the most suitable TTS model for their specific use case and provides valuable guidance for future TTS research and development.
>
> **Q2. Clarify how the Win-Rate metric used in this study correlates not only with MOS but also with objective measures such as word error rate (WER) and PESQ.**
>
> We thank the reviewer for this valuable suggestion.
>
> Our analysis shows a strong, statistically significant positive correlation (Spearman: 77.00%, p < 0.0001) between our overall win-rate and negative WER, validating that our metric aligns with objective intelligibility. We use the negative WER so that higher values indicate better performance, in line with the interpretation of win rate. The strength of this correlation varies informatively across categories, as shown below.
>
> | Category | Spearman Correlation (Win-Rate vs. -WER) |
> | :--- | :--- |
> | Foreign Words | 86.82% |
> | Paralinguistics | 71.38% |
> | Syntactic Complexity | 67.09% |
> | Pronunciation | 59.13% |
> | Questions | 39.86% |
> | Emotions | 35.77% |
>
> The correlation is strongest in challenging domains like Foreign Words and Paralinguistics, where weaker models are prone to synthesis failures that produce a high WER and are correctly penalized by our judge. Conversely, it is weaker in categories like Emotions, where a model can be lexically perfect (low WER) but still receive a low win-rate for failing to convey the required prosody, a nuance our benchmark is designed to capture.
>
> We considered using the baseline TTS model's output as a reference for PESQ calculation, but determined this approach to be methodologically inappropriate. PESQ is intended to assess signal degradation relative to a ground-truth reference, and a TTS-generated baseline audio cannot be treated as the ground-truth. Furthermore, the metric would heavily penalize differences in speaker identity and prosodic interpretation between the two models, making the score uninterpretable and not reflective of synthesis quality. We therefore chose to rely on metrics appropriate for the TTS task: WER for intelligibility and our proposed win-rate metric, which has high correlation with real human preference.
>
> References:
>
> [1] HuggingFace: "sesame/csm-1b".
>
> [2] HuggingFace: "canopylabs/orpheus-3b-0.1-ft".
>
> [3] Xu, Jin, et al. "Qwen2.5-Omni Technical Report" arXiv preprint arXiv:2503.20215 (2025).
>
> [4] Kim, Jaehyeon, et al. "Conditional Variational Autoencoder with Adversarial Learning for End-to-End Text-to-Speech" arXiv preprint arXiv:2106.06103 (2021).
>
> [5] HuggingFace: "hexgrad/Kokoro-82M".
>
> [6] HuggingFace: "kyutai/tts-1.6b-en_fr".
>
> [7] Chen, Yushen, et al. "F5-TTS: A Fairytaler that Fakes Fluent and Faithful Speech with Flow Matching" arXiv preprint arXiv:2410.06885 (2025).
>
> [8] Yamagishi, Junichi, et al. "CSTR VCTK Corpus: English Multi-speaker Corpus for CSTR Voice Cloning Toolkit". University of Edinburgh. The Centre for Speech Technology Research (CSTR) (2019).
>
> [9] He, Haorui, et al. "Emilia: An extensive, multilingual, and diverse speech dataset for large-scale speech generation" arXiv preprint arXiv:2407.05361 (2024).

---

### Official Review · Reviewer_oEok · 2025-07-05

**Rating:** 4
**Confidence:** 3

**Summary:**

The paper introduces EmergentTTS-Eval, an automated TTS benchmark covering six challenging scenarios (emotions, paralinguistics, foreign words, syntax, complex pronunciation, and questions), using LLM-augmented prompts to generate 1,645 diverse test samples. It proposes a model-as-a-judge approach with a Large Audio Language Model (LALM) to assess TTS outputs across multiple dimensions (emotion, prosody, intonation, pronunciation), showing strong correlation with human judgments. Evaluations of leading TTS systems (e.g., 11Labs, Deepgram, OpenAI) reveal fine-grained performance gaps, with the benchmark and evaluation code released openly for reproducibility.

**Dataset Code Accessibility:**

Yes

**Ethical Considerations:**

No, there are no or only very minor ethics concerns

**Final Justification:**

Thank you to the authors for thoroughly addressing all the concerns raised. I have carefully read through all the reviews and the rebuttal. Overall, I found that the authors have responded well to the points raised. Therefore, I am inclined to raise my score.

**Limitations Weaknesses:**

1. Presentation and Readability Issues:
The organization and writing of this paper require further refinement. Specifically:

   Figures 3 & 4: The font size is too small, making the text difficult to read.

   Tables 1 & 2: Restructuring is recommended to improve clarity and presentation.

   Table 3(b) & Figure 4 titles: The spacing is insufficient, which may lead to misinterpretation.

2. Methodological Concerns:
As acknowledged in the Limitations section, the current evaluation pipeline struggles with multilingual scenarios. However, the proposed benchmark considers "foreign words" as a key dimension—an apparent contradiction. This inconsistency raises concerns about the pipeline’s reliability in real-world applications.

**Strengths Contributions:**

This paper is well-motivated, focusing on a critical gap in current TTS evaluation benchmarks: the evaluation for more sophisticated semantic understanding and finer-grained expressive capabilities.

---

> ### Author Rebuttal · Authors · 2025-07-31
>
> We thank the reviewer for recognizing the gap in current TTS evaluation methods (i.e., lack of tools for assessing more sophisticated semantic understanding and finer-grained expressive capabilities), and for acknowledging how our benchmark fills it. We address the specific questions below.
>
> **Q1: ...font size is too small...spacing is insufficient...**
>
> We will resolve these formatting issues in the camera-ready version.
>
> **Q2: ...Limitation section...struggles with multilingual scenarios...foreign words...**
>
> Thanks for the comment. We think the reviewer misunderstood the limitation we mentioned in Section 5, specifically the sentence "our multilingual evaluation focuses on Latin transcriptions rather than native character sets, which doesn’t fully capture the challenges of true multilingual TTS.". We have already provided an explanation for our deliberate design of foreign words category in Appendix A.2, but would welcome this opportunity to expand on that.
>
> Our work draws a clear distinction between two scenarios: (i) monolingual speech generation from native scripts in non-English languages, and (ii) rapid code-switching between two languages using the Latin script. We focus on the latter, more challenging scenario, and constructed samples that contain the ten most spoken languages worldwide excluding English (Mandarin Chinese, Hindi, Spanish, French, Arabic, Russian, Portuguese, Japanese, German, and Indonesian). In terms of evaluation, code-switching poses greater challenges than monolingual speech generation in a non-English language. Our framework evaluates not only the model's pronunciation but also its ability to maintain natural prosody across language boundaries and to infer phonetics correctly from context. Standard metrics like WER, while effective for monolingual speech in native scripts, fall short in this area. To isolate the evaluation of TTS capabilities from the complexities of script recognition, we present all non-English words in standard Latin characters.
>
> Furthermore, the limitation we mentioned is not about the evaluation framework, but about the scope we chose to address in this paper. Our EmergentTTS-Eval framework is designed to be extensible and can support additional languages by incorporating new categories of text and LALM judge prompts. However, each language may require special design of the judge prompts to accurately evaluate complex prosody, expressiveness, and linguistic nuance. For instance, evaluating Chinese should involve tonal accuracy in challenging cases such as tongue-twisters and adherence to the rhythmic structure of classical poetry; German evaluation should address prosodic stress in long compound nouns; and French should consider correct pronunciation across word boundaries (liaison). These extensions are beyond the current scope and are left for future work, which our framework is well-positioned to support.

---

> > ### Comment · Reviewer_oEok · 2025-08-06
> > **Official comments by Reviewer oEok**
> >
> > Thank you to the authors for thoroughly addressing all the concerns raised. I have carefully read through all the reviews and the rebuttal. Overall, I found that the authors have responded well to the points raised. Therefore, I am inclined to raise my score.

---

### Decision · Program_Chairs · 2025-09-18

**Decision:**

Accept (poster)

**Comment:**

This paper introduces a benchmark for a range of TTS scenarios, with both test sample generation and evaluation showing high correlation with humans. There is strong consensus among all reviewers that the work is well-motivated given the lack of more granular expressive capability measurement in existing TTS benchmarks (a gap which this benchmark fills), and the lack of scalability of existing gold standard metrics like MOS (towards which the authors instead propose a LALM evaluation framework). Concerns about the paper were more than adequately addressed during the discussion phase: e.g., with regard to multilingual scenarios (by clarifying the code-switching scenario), the types of TTS models on which evaluation can be applied (for which the authors performed additional experiments on a range of models), the correlation between the proposed Win-Rate metric and other standard metrics like WER (for which the authors calculated the correlations), insight on the performance of LLM judges (for which the authors provided qualitative and quantitative evidence), and insight on the divergence between MOS and win-rate (authors may want to include their clarifying discussion period responses to reviewer v3gf in the paper). Overall, if the above points are incorporated into the paper, this is extremely compelling work that can allow for much more streamlined benchmarking of TTS models.